# Accuracy and Inter-Analyst Agreement of Visually Estimated Sea Ice Concentrations in Canadian Ice Service Ice Charts Using HH SAR

Angela Cheng[1, 2], Barbara Casati[3], Adrienne Tivy[1], Tom Zagon[1], Jean-François Lemieux[3], and  L. Bruno Tremblay[2]

[1]Canadian Ice Service, Environment and Climate Change Canada, Ottawa, Ontario, Canada
[2]Department of Atmospheric and Oceanic Sciences, McGill University, Montréal, Québec, Canada
[3]Environnement et Changement Climatique Canada, Recherche en Prévision Numérique Environnementale, Dorval, Québec, Canada

**Correspondence:** A. Cheng, 719 Heron Road, Ottawa, Ontario, K1A 0H3, Canada. (angela.cheng@canada.ca)

**Abstract.** This study compares the accuracy of visually estimated ice concentrations by eight analysts at the Canadian Ice Service against three standards: i) ice concentrations calculated from automated image segmentation, ii) ice concentrations calculated from automated image segmentation that were validated by the analysts, and iii) the modal ice concentration estimate by the group. A total of 76 pre-defined areas in 67 RADARSAT-2 images are used in this study. Analysts over-estimate ice concentrations when compared to all three standards, most notably for low ice concentrations (1/10 - 3/10). The spread of ice concentration estimates is highest for middle concentrations (5/10, 6/10), and smallest for 9/10. The over-estimation in low concentrations and high variability in middle concentrations introduces uncertainty in the ice concentration distribution in ice charts. The uncertainty may have downstream implications for numerical modeling and sea ice climatology. Inter-analyst agreement is also measured to determine which classifier's ice concentration estimates (analyst or automated image segmentation) disagreed the most. It was found that one of the eight analysts disagreed the most, followed second by the automated segmentation algorithm. This suggests high agreement in ice concentration estimates between analysts at the Canadian Ice Service. The high agreement, but consistent overestimation, results in an overall accuracy of ice concentration estimates in polygons to be 39%, 95% CI [34%, 43%] for an exact match in ice concentration estimate with calculated ice concentration from segmentation, and 84%, 95% CI [80%, 87%] for +/- one ice concentration category. Only images with high contrast between ice and open water, and well-defined floes are used: true accuracy is expected to be lower than what is found in this study.

## 1   Introduction

Sea ice charts are routinely made by national Ice Services to provide accurate and timely information about sea ice conditions. These charts are produced to support navigation in polar regions, to provide information to local communities and to monitor the long-term evolution of sea ice conditions (i.e, climatology). Ice analysts and forecasters generate these charts by using various data sources, including remotely sensed imagery, to quantify various sea ice characteristics, including sea ice concentration.

Analysts and forecasters at the Canadian Ice Service (CIS) predominantly rely on RADARSAT-2 imagery for monitoring sea ice conditions. Analysts at the CIS identify areas with similar ice conditions and open water for navigational purposes, then manually delineate them with polygons. The analyst then assigns an estimated concentration value for the polygon using the visual segmentation. The ice concentration is expressed in categories, as a percentage rounded to the nearest tenth (i.e. 1/10, 2/10, etc).

A number of studies have been done to develop, assess, or improve upon algorithms for automated calculation of ice concentration from remotely sensed images. Algorithms have been built for automated ice concentration retrieval using different sensors. The NASA Team and Bootstrap algorithms use Special Sensor Microwave/Imager, (commonly referred to as SSM/I), although other algorithms exist for AVHRR and RADARSAT-2 SAR images (e.g., Belchansky and Douglas, 2002; Williams et al., 2002; Meier, 2005; Hebert et al., 2015; Scott et al., 2012). The process of automatically calculating ice concentration from remotely sensed images requires classifying each pixel of the image into a category. In the simplest case, the categories are ice and open water. In more complex cases, the ice is categorized by type, such as multi-year ice or landfast ice, or by thickness (e.g. thick vs. thin). Classified pixels are then grouped by category to produce segmentation results. The sea ice concentration can then be derived for a given area by taking the sum of the total number of sea ice pixels and dividing by the total number of pixels.

Manually derived products from SAR are assumed to be the most accurate source of information on ice concentration by many users (Karvonen et al., 2005). Therefore, many studies rely on the manually-derived ice products as a ground truth when developing automated ice retrieval techniques (Komarov and Buehner, 2017; Karvonen, 2017). However, due to a limited number of studies, the accuracy of manually derived ice concentrations is not well understood. Karvonen et al. (2015) conducted a study in an attempt to quantify this; they compared the sea ice concentrations assigned by five separate working groups (containing up to five ice analysts each) for 48 polygons predefined by an ice analyst using two high resolution ScanSAR images in the Baltic Sea. They found deviation in ice concentration estimates between ice analysts, some of which were significant, especially for polygons in mid-range ice concentrations. However, their study was geographically and temporally limited to only two SAR images in the Baltic Sea. Other than this study, there has been little measure of the spread and variability of sea ice concentration estimates by human analysts, or the subsequent uncertainty of ice charts produced by operational ice analysts.

The uncertainty of sea ice concentration estimates can result in downstream uncertainties for applications that rely on sea ice charts. For example, sea ice concentration estimates from Canadian Ice Service charts are used as a data source for input to initialize sea ice models (Smith et al., 2016; Lemieux et al., 2016). The error in the initial condition of sea ice concentration estimates can propagate and grow with time, and impact the accuracy of predictions from numerical models (Parkinson et al., 2001). Uncertainty of ice concentration estimates could also impact the accuracy of climatology studies of ice concentration derived from operational ice charts, although that has not been investigated.

The main objective of this study was to determine the probability that a given ice concentration in a Canadian Ice Service ice chart polygon reflects the ice concentration found in the corresponding SAR image used by analysts to create the ice chart. To achieve this, we assessed:

(i) How accurate analysts and forecasters are at visually estimating ice concentration when compared to calculated ice concentration from image segmentation under best case scenarios (i.e. image segmentation adequately resembles the visual segmentation done by an analyst or forecaster)

(ii) How consistent analysts and forecasters are with one another in visually estimating ice concentration from SAR imagery.

The paper is structured as follows. Section 2 describes the data and standards for ice chart creation. Section 3 describes the methodology for generating the sample polygons used in this study, calculating total ice concentration from image segmentation, and capturing analyst estimates of ice concentration. Section 4 describes the producer and user's accuracy, as well as the two skill scores used in this study. Section 5 provides the results of the comparison between visually estimated ice concentrations and calculated ice concentrations using the skill scores. Section 6 compares visually estimated ice concentrations against

the modal ice concentration value. Section 7 describes the accuracy of visually estimated ice concentrations in polygons. The paper concludes with a discussion in the final section.

## 2   Ice Charting

Section 2 describes elements of ice charting. Section 2.1 describes the remote sensing data that is the primary input data source for generating ice charts. Section 2.2 briefly describes type of ice charts created at the Canadian Ice Service. Section 2.3 gives

an overview of the egg code, which is the international standard used for ice charting, and is the method for communicating ice concentrations in an ice chart.

### 2.1   Remote Sensing for Monitoring Ice Concentration

Sea ice is routinely monitored by Ice Services using satellites due to their ability to acquire images covering large spatial areas. Passive microwave and synthetic aperture radar (SAR) sensors are preferred over optical imagery because of their ability to

see through clouds. Optical satellites rely on solar illumination, which is absent in Arctic regions during polar night. Passive microwave observations often have coarse resolution (i.e. 50 kilometer) whereas SAR data can be acquired in high (i.e. 50 meter) resolution. Low spatial resolution makes it difficult to resolve sea ice conditions in certain conditions or geographic areas. SAR provides consistent, high resolution coverage of the Arctic without cloud interference or limitations due to lack of solar illumination.

The CIS relied on RADARSAT-1, a SAR sensor, for ice charting beginning in 1996 until its decommissioning in 2013. The Canadian Ice Service currently relies predominantly on RADARSAT-2, but will start to use the RADARSAT Constellation Mission (RCM) operationally, following its recent launch in 2019. In the 2017 calendar year, the Canadian Ice Service received approximately 45 000 SAR scenes between Sentinel-1 and RADARSAT-2, and another 85 000 scenes from various satellites including GOES, MODIS, AMSR, and VIIRS. The lower number of SAR scenes reflects the fact that RADARSAT scenes

are geographically targeted acquisitions ordered by the CIS, while GOES, MODIS, AMSR and VIIRS are publicly available

swaths acquired for general use. The latter are less targeted for CIS Operations, but useful as a secondary, supplemental data source.

Ice Services have had difficulty implementing systems to automate sea ice interpretation from satellite imagery. Automated calculation of sea ice concentration requires first classifying the pixels in an image in to categories of ice (i.e. first year ice, multi-year ice, etc.) or open water, then calculating the proportion of ice within a given area. Automated sea ice algorithms in SAR rely on interpretation of sea ice backscatter, which can be ambiguous (Zakhvatkina et al., 2019). For example, open water under low wind conditions yield similar backscatter to first year ice, making it difficult to distinguish automatically. During the melt season, sea ice forms a layer of meltwater on top of the sea ice, which can yield similar backscatter values to that of open water–confusing algorithms to classify it as open water. Many attempts have been made to automatically classify sea ice in SAR scenes using a variety of methods but have had difficulty in conditions such as those previously described (Zakhvatkina et al., 2019). In many of these cases, expert analysts are able to detect sea ice where the algorithms cannot.

Polarizations have provided additional data that have been useful for implementing automated sea ice classification. Polarization refers to the orientation of the electromagnetic waves sent and received by the sensor. The main polarizations used for sea ice monitoring with RADARSAT-2 are (1) horizontal transmit and horizontal receive (HH), and (2) horizontal transmit and vertical receive (HV). The HV band has been shown to be less sensitive to the incidence angle of the satellite. The combination of both HH and HV channels together has been shown to better distinguish between sea ice concentrations than either channel alone (Karvonen, 2014). On the other hand, only the HH polarization was available for sea ice monitoring with RADARSAT-1. RCM will provide compact polarimetry modes, which will provide additional information. Automation of sea ice classification algorithms currently use dual-polarization imagery, but will use compact polarimetry as it becomes available.

Despite technological advances in SAR satellites and a long history of development of automated techniques, Ice Services still continue to rely on manually drawn ice charts to identify sea ice conditions because automation of sea ice classification has significant limitations (i.e. difficulty in separating multi-year ice from first year ice during summer melt). Furthermore, analysts are able to provide additional information that automated classification cannot (i.e. analysts can provide total concentration, partial concentration, stage of development, etc, in a single chart). However, ice charting is time-consuming, and the number of images acquired by satellites is expected to increase. Therefore, development of automated sea ice classification algorithms for ice charting continues to be a pressing need for operational Ice Services.

## 2.2 Chart Types

A number of different types of charts are generated at the Canadian Ice Service (e.g. regionals, dailies, image analyses, concentration, stage of development, etc.), which vary due to the chart's purpose, relevant time, or underlying data sources (Canadian Ice Service); (refer to Dedrick et al. (2001) on National Ice Center charts for the U.S.). Image analysis charts are created by visually interpreting specific satellite images. These charts are constrained to the geographic extent and resolution of the corresponding satellite images. Daily ice charts combine different sources of information, introducing variability between the ice chart and the satellite image.

## 2.3 Egg Code

The egg code is a World Meteorological Organization international standard for coding ice information (see Figure 1), (WMO, 2014). Each polygon in an ice chart is assigned an egg code with corresponding values. The egg code contains information on the ice concentration (C), stage of development (S), and the predominant form (F) of ice (floe size), within an oval shape. The top value in the egg code is the total ice concentration, which includes all stages of development of ice. Total ice concentration are expressed in categories, where the ice concentration as a percentage is rounded to the nearest tenth. Less than 1/10 of sea ice is used to denote open water, which is not the absolute absence of ice but is the definition of ice less than 1/10. Partial concentration is used when more than one ice type is present within the delineated polygon. No partial concentration is reported when only one ice type is found. In our study, we only considered total concentration rather than partial concentrations.

## 3 Ice Concentration Estimates

In this paper we consider three standards against which analysts' visually estimated ice concentration is compared against, due to the absence of absolute ground truth. The standards used are: i) ice concentrations derived from automated segmentation; ii) ice concentrations derived from automated segmentation that have been validated by analysts; and iii) the mode of ice concentration estimates given by analysts. This section describes the methodology for creating sample polygons used in this study, calculating ice concentrations from automated segmentation, and capturing visual estimation of ice concentration from participating ice analysts.

### 3.1 Case Studies Selection and Polygon Definition

RADARSAT-2 images were randomly selected from the Canadian Ice Service image archive. Each image was manually reviewed to find areas of clear contrast between water and ice to optimize segmentation capability and reduce potential ambiguity in visual analysis.

A former operational analyst delineated potential polygons for the sample of selected cases used in this study. Polygons were drawn in areas with high contrast between ice and open water (to optimize the algorithm's ability to differentiate between ice and water) and areas of fractional ice cover (since there is little value in evaluating analysts' ability to estimate 0/10 or 10/10 ice concentration). Unlike a traditional ice chart where analysts segment entire images in to polygons; we only drew polygons in areas of interest.

The images used for this study were selectively picked to be areas with well-defined floes with high contrast against the black (water) background. SAR image quality varies from image to image, and even within image. Likewise, the structure of sea ice in Canadian waters can vary greatly, with brash and rubble ice along the East Coast and well-defined floes in the Beaufort Sea. Ice without well-defined spatial structure may not be captured due to the resolution of the sensor. For example, first year ice can appear similar to open water, making it difficult to determine its edges. Brash ice is composed of small pieces of ice (less than 2 meters in diameter) that cannot be resolved at the resolution of the (SAR) sensor. Furthermore, segmentation

of sea ice in visually ambiguous conditions (i.e. first year ice during the melt season; brash ice; etc.) by automated algorithms is still sub-optimal. As a result, we did not present analysts with ice conditions that would have been difficult to automatically segment. The sea ice types used in the samples of this study are not representative of all sea ice conditions typically found in Canadian Service Ice Charts. This study quantifies the accuracy of sea ice concentration estimates under the best case scenario of well-defined floes in very clear SAR images. It is expected that accuracy would decrease under brash ice conditions and/or poor image quality."

We assessed if there were differences in the size of polygons drawn for this study and the sizes of polygons in published charts since polygon sizes could impact analyst ability to estimate ice concentration. The polygon sizes were compared to polygon sizes from two types of published operational charts: daily charts and image analyses. The image analyses and daily charts used the same RADARSAT images that were used to delineate polygons used in this study. Since the polygons were delineated differently, sometimes the sample polygon would spatially intersect with two or more polygons, making it difficult to directly compare the sizes of polygons. We addressed this by identifying the polygon with the greatest spatial intersection with the sample polygon, and comparing the two areas. Figure 2 shows the difference between polygon sizes. Polygon sizes were not normally distributed. Under a Wilcoxon-Mann-Whitney rank test, polygons from image analyses and daily charts are not significantly different in their sizes (p = 0.226). On the other hand, polygons generated for this study are significantly smaller than polygons from image analysis charts (p = 0.002), although there is overlap in the size range. Polygons generated for this study are also significantly smaller than polygons from daily charts (p = 0.071), with less overlap in the size range than for the image analyses.

### 3.2 Ice Concentration Estimates Using Automated Image Segmentation

The University of Waterloo MAp Guided Ice Classification (MAGIC) system was used to classify RADARSAT-2 pixels as ice or open water (Clausi et al., 2010). MAGIC uses an Iterative Region Growing using Semantics (IRGS) framework to classify pixels into categories. The RADARSAT-2 images analyzed in this study were run through the MAGIC algorithm using default parameters. As an input to MAGIC, we specified only two classes in the polygon to force MAGIC to segment the pixels into ice or open water only. After running MAGIC we performed a visual inspection (and when necessary, we manually assigned the classification of ice or open water) to ensure that the resulting ice concentration was calculated correctly. Figure 3 shows an example of the output from MAGIC.

Only the HH band was used for both segmentation and visual interpretation in this study. Typically, ice charting is done with HH as a primary polarization, and HV is only used to distinguish ambiguous ice types. However, the sample polygons used in this study focused on examples with minimal ambiguity.

The total ice concentration was calculated as a ratio of the ice pixels to the total number of pixels in the polygon,

$$C = \frac{N}{T} \tag{1}$$

where $N$ is the number of ice pixels, and $T$ is the total number of pixels in the polygon. Result values were binned into categories to reflect the ice concentration categories used in operational ice charts (Table 1).

**Table 1.** Conversion of $C$ to Concentration Categories $C_t$ found in ice charts

| Percentage | Category |
|:---:|:---:|
| 0.00 - 0.10 | 0 |
| 0.10 - 0.15 | 1 |
| 0.15 - 0.25 | 2 |
| 0.25 - 0.35 | 3 |
| 0.35 - 0.45 | 4 |
| 0.45 - 0.55 | 5 |
| 0.55 - 0.65 | 6 |
| 0.65 - 0.75 | 7 |
| 0.75 - 0.85 | 8 |
| 0.85 - 0.95 | 9 |
| 0.95 - 0.99 | 9+ |
| 1.00 | 10 |

### 3.3 Ice Concentration Estimates From Operational Analysts

A total of eight analysts and forecasters were given a customized user interface designed for this study (Figures 4, 5). (For the remainder of this paper, the term analyst will be used to indicate analyst or forecaster). Two sample polygons were used as a test run to ensure that analysts were familiar with the user interface before the assessment. After completing the test run, analysts completed the following sequence for each polygon presented:

(i) Input an estimated ice concentration value for the delineated area. Options were restricted to only the values found in the standard ice-chart egg code.

(ii) The analyst was presented with the segmentation results from MAGIC only after submitting the value in the previous step. The analysts were able to toggle back and forth between the original image and the segmentation results.

(iii) The analyst was then asked if they agreed or disagreed with the segmentation results.

(iv) If the analyst input "Disagree," they were then asked to state if they felt the segmentation algorithm over-estimated or under-estimated the ice.

(v) Analysts were asked to input any additional comments. This allowed for comments to explain why they felt their estimate differed from the segmentation results.

The analyst repeated these five steps for all polygons in random order until the entire set was completed. There was a total of 76 polygons analyzed by eight analysts, which resulted in a total of 608 cases. Each polygon had a total of eight responses

as all analysts completed the same set of polygons. We chose to present the segmentation results *after* analysts had input an estimated ice concentration value in order to prevent potentially biasing results. We considered the cases where the analyst stated "agree" in step (iii) as valid; cases where analyst stated "disagree" in step (iii) were considered invalid. This was done to allow us to subset the ice concentrations from MAGIC to only those cases where analysts found the segmentation was valid.

## 4 Accuracy and Agreement Skill Scores

This section describes the two skill scores and measures used in this study to determine accuracy and agreement. We use an error matrix, producer's accuracy, user's accuracy, and the kappa statistic for assessing the accuracy of classifiers in remotely sensed imagery (Lillesand et al., 2015). The same statistical framework is known in verification as the multi-categorical contingency table with its calibration-refinement factorization, likelihood-base rate factorization, and Heidke skill score (Murphy and Winkler, 1987; Wilks, 2011; Joliffe and Stephenson, 2012). We employ these measures in our study to assess the accuracy of ice concentration estimates provided by the analysts compared to ice concentrations calculated from the automated image segmentation. In addition to the kappa statistic, we use Krippendorff's alpha to measure agreement between analysts. This measure is often used in counseling, survey research, and communication studies to measure inter-rater reliability–that is, the level of agreement between multiple judges (Hallgren, 2012). Whereas the kappa statistic is restricted to only comparing two judges, Krippendorff's alpha can compare multiple individuals. In our study, it was used to measure the agreement between individual analysts, and to identify the level of disagreeent between analyst and MAGIC. In the context of this paper, we refer to it as inter-rater agreement rather than inter-rater reliability, (so as not to be confused with reliability in the verification context).

### 4.1 Multi-categorical Contingency Table, Producer's Accuracy, and User's Accuracy

The multi-categorical contingency table was used to compare the analyst's visual estimation of ice concentration against the segmentation results. A total of 12 ice concentration categories were possible, producing a 12x12 matrix of possibilities. The entries in this matrix are normalized by the total counts (608) and correspond to the joint probabilities $p_{m,x} = p(m, x)$ of ice concentrations $m$ and $x$ (as estimated by MAGIC and the analysts). Perfect accuracy is achieved when all possible comparisons lie on the diagonal of the multi-categorical contingency table. The reference dataset is typically on the horizontal axis, and the classifier on the vertical. In our case, we assigned analyst estimates along the vertical and used the ice concentrations derived from MAGIC along the horizontal.

First, we define

$$
m = \begin{cases}
0, & \text{if MAGIC calculated 0/10 ice concentration} \\
1, & \text{if MAGIC calculated 1/10 ice concentration} \\
\vdots & \qquad\qquad \vdots \\
9, & \text{if MAGIC calculated 9/10 ice concentration} \\
10, & \text{if MAGIC calculated 9+/10 ice concentration} \\
11, & \text{if MAGIC calculated 10/10 ice concentration}
\end{cases}
$$

and

$$
x = \begin{cases}
0, & \text{if the analyst estimated 0/10 ice concentration} \\
1, & \text{if the analyst estimated 1/10 ice concentration} \\
\vdots & \qquad\qquad \vdots \\
9, & \text{if the analyst estimated 9/10 ice concentration} \\
10, & \text{if the analyst estimated 9+/10 ice concentration} \\
11, & \text{if the analyst estimated 10/10 ice concentration}
\end{cases}
$$

For each ice concentration category $m$ estimated by MAGIC, the producer's accuracy $p(x|m)$ is the proportion of time that the analyst estimates the same ice concentration category.

Symmetrically, for each ice concentration category $x$ estimated by the analyst, the user's accuracy $p(m|x)$ is the probability that MAGIC will report the same value as the analyst. In the Murphy and Winkler (1987) verification framework, the user's accuracy is the conditional probability of MAGIC estimating a specific ice category, given that the analyst estimated the same category. The user's accuracy is computed by dividing the number of correctly estimated ice concentrations by the row total for each category (the row totals correspond to the marginal probabilities of the analysts estimates).

The joint probability $p(x, m)$, of MAGIC estimating the $m$ sea-ice category and the analyst estimating the $x$ sea-ice category, informs on the accuracy (aka agreement between analyst and MAGIC). Best accuracy is achieved when $p(x, m) = 1$ for $x = m$ (along the diagonal) and $p(x, m) = 1$ for $x \neq m$ (off the diagonal). The joint probability $p(x, m)$ is related to both the producer's and user's accuracy, as:

$$
p(m, x) = p(m|x)p(x) = p(x|m)p(m) \tag{2}
$$

where $p(x)$ and $p(m)$ are the marginal distributions of analyst and MAGIC estimated ice concentration categories $x$ and $m$, respectively. User's and producer's accuracy, weighted by the marginal frequencies, are indicators of the overall MAGIC and analyst agreement.

## 4.2 Kappa Statistic

The kappa statistic is a skill score which measures how well an analyst can perform compared to chance. In forecast verification the kappa statistics in known as the Heidke Skill Score, and it is the skill score constructed from the percent correct against

random chance (Wilks, 2011; Joliffe and Stephenson, 2012). The kappa statistic can be calculated for any contingency table to measure the level of agreement between analysts and the segmentation algorithm. This measure takes into account the possibility of chance agreement between analysts and MAGIC when determining the agreement found between them.

The kappa statistic, $\kappa$, is calculated as

$$\kappa = \frac{p_0 - p_e}{1 - p_e} \tag{3}$$

where $p_0$ is the agreement between the analyst and the segmentation results, $p_e$ is the agreement that a random estimation is expected to achieve, and 1 is the value attained by $p_0$ when there is perfect agreement.

The observed agreement $p_0$ (known also as percent correct), is the sum of the diagonal joint probabilities

$$p_0 = \sum_{i=1}^{k} p_{ii} \tag{4}$$

for the $k$ ice concentration categories.

The expected chance agreement, $p_e$, is

$$p_e = \sum_{i=1}^{k} p_i^h p_i^v \tag{5}$$

for the $k$ ice concentration categories, where $p_i^h$ is the marginal probability for the $i^{th}$ row, and $p_j^v$ is the marginal probability for the $j^{th}$ column. The product of the marginal probabilities $p_i^h p_j^v$ gives the joint probability for the categories i and j occurring at the same time by chance, in virtue of the Bayes Theorem.

A weighted kappa can be used to apply a penalty to disagreements which increases with distance from the diagonal. This is unlike the unweighted case above, where all disagreements are equally penalized. In this study, we used a linearly weighted kappa (for each ice concentration category away from the diagonal, we increase the penalty by one).

A $\kappa$ value of 1 indicates complete agreement between the estimated ice concentration and the calculated ice concentration from image segmentation. A $\kappa$ value close to 0 indicates agreement close to that expected by chance. A negative value is theoretically possible, which indicates that an analyst is worse than random chance. However, negative values are rare, and 0 is often used as a lower bound. Landis and Koch (1977) suggested values greater than 0.8 represent strong agreement, values between 0.6 and 0.8 represent moderate agreement, values between 0.4 and 0.6 represent mild agreement, and values below 0.4 as poor agreement.

### 4.3 Krippendorff's Alpha

Krippendorff's alpha is a skill score that measures the level of agreement between multiple analysts. Krippendorff's alpha ranges from 0 to 1, where 1 indicates perfect agreement. Krippendorff suggests that $\alpha = 0.8$ indicates good skill, although he

**Table 2.** Reliability data matrix. This table is used to create the coincident matrix.

| polygons $u$ | 1 | 2 | ... | $u$ | ... | $N$ |
|---|---|---|---|---|---|---|
| Analyst 1 | $c_{11}$ | $c_{12}$ | ... | $c_{1u}$ | ... | $c_{1N}$ |
| $i$ | $c_{i1}$ | $c_{i2}$ | ... | $c_{iu}$ | ... | $c_{iN}$ |
| $j$ | $c_{j1}$ | $c_{j2}$ | ... | $c_{ju}$ | ... | $c_{jN}$ |
| . | . | . | ... | . | ... | . |
| $m$ | $c_{m1}$ | $c_{m2}$ | ... | $c_{mu}$ | ... | $c_{mN}$ |
| Number of analysts | $m_1$ | $m_2$ | ... | $m_u$ | ... | $m_N$ |

suggests a value of $\alpha = 0.667$ as a tentatively acceptable lower limit. Krippendorff's alpha, $\alpha$, is calculated as

$$\alpha = 1 - \frac{d_o}{d_e} \tag{6}$$

where $d_o$ is the observed disagreement, and $d_e$ is the expected disagreement (when there is no reliability).

The observed disagreement, $d_o$ is

$$d_o = \frac{1}{n} \sum_c \sum_k o_{ck} \delta_{ck}^2 \tag{7}$$

and the expected disagreement, $d_e$ is

$$d_e = \frac{1}{n(n-1)} \sum_c \sum_k n_c n_k \delta_{ck}^2 \tag{8}$$

The value $n$ is the total number of pairs of values $c$ and $k$. The values $o_{ck}$, $n_c$, $n_k$, and $n$ are all frequencies in the coincident matrix defined in Table 3, which is built from the reliability matrix (Table 2). The coincident matrix is

$$o_{ck} = \sum_u \sum_{i=1}^m \sum_{j \neq i} \frac{1}{m_u - 1} \text{ iff } c_{iu} = c, c_{ju} = k, \text{ and } m_u \geq 2 \tag{9}$$

For this study, the ice concentration categories were treated as ordinal data, which is applicable as the ice concentration categories can be treated as ranks. That is, the lowest rank has the least amount of sea ice and the highest rank has the most. For ordinal data, the metric difference, $\delta_{ck}^2$, is

$$\delta_{ck}^2 = (\sum_c^k n - \frac{n_c + n_k}{2})^2 \tag{10}$$

## 5  Comparison of Estimates from Generated Polygons against MAGIC

The first objective of the study was to compare analyst estimated ice concentrations with ice concentrations derived from image segmentation using MAGIC.

**Table 3.** Coincident matrix. The coincident matrix is built by counting the frequencies of $ck$ pairs occurring in the reliability data matrix (Table 2).

| Values | 1 | $k$ | ... | Row Total |
|--------|-----|------|-----|-----------|
| 1 | $o_{11}$ | $o_{1k}$ | ... | $n_1$ |
| . | . | . | ... | . |
| $c$ | $o_{c1}$ | $o_{ck}$ | ... | $n_c = \sum_k o_{ck}$ |
| . | . | . | ... | . |
| Column Total | $n_1$ | $n_k$ | ... | $n = \sum_c \sum_k n_{ck}$ |

Automated sea ice classification algorithms often use sea ice charts as a truth dataset for verification since they cover large geographic areas and have been produced for many years by many Ice Services. Furthermore, while there are differences among ice charts, they generally agree with respect to types of ice present and where they occur. Therefore, we compared the visually estimated ice concentrations against all segmentation results first. Next, we compared the visually estimated ice concentrations against segmentation results only for the cases where the analyst determined the segmentation result was valid. This was done by asking analysts, as part of the work flow, whether they felt the segmentation adequately captured the ice conditions in each image. Of the 76 polygons analyzed by the eight analysts, there were 181 times where the analyst disagreed with the image segmentation results. Of the 181 disagreements, only six were reported to have over-estimated sea ice by MAGIC. The remaining 175 disagreements were reported to have under-estimated sea ice concentration by MAGIC. Furthermore, the accepted segmentation results varied between analysts. In 36.8% of total polygons, the analysts were unanimous in agreement with the outcome of the automatic segmentation. The remaining polygons had lower levels of acceptance by analysts. There were no polygons in the study where all analysts found the results unacceptable (Figure 6).

We first considered analyst responses against segmentation results regardless of the validity of the segmentation (validity determined by the analyst). Figure 7 shows estimates from the analysts (all individuals combined together) against segmentation results. Perfect agreement between analyst estimation and the segmentation results lie along the diagonal; entries above (below) the diagonal show over (under) estimation by analysts: the analysts tend to over-estimate the ice category with respect to MAGIC (Figure 7). Figure 8 shows the individual contingency tables of responses for each analyst that participated in this exercise.

Figure 9 compares the two marginal distributions in Figure 7. Over-estimation of all ice concentrations resulted in an increase in the number of polygons with high ice concentration (9/10 to 10/10).

We then subset the data down to only those responses where the analyst stated the segmentation was valid. Figure 10 shows the combined responses from all participants in this study. Individual responses are shown in Figure 11. As expected, removing the cases where analysts reported the segmentation results were invalid reduced the bias and narrowed the spread

of the analysts' ice concentration estimates (showing that the analysts' estimates meet consistency), although a bias towards overestimation persisted.

Comparison of the marginal distribution of the MAGIC estimates for the whole sample of cases (bottom panel of Figure 7), versus the one for the subset of cases in which analysts agreed with the automated segmentation (bottom panel of Figure 10), show qualitatively that most of the disagreement occurred for low-concentration values (2/10,3/10,4/10) and that the category 8/10 exhibits the largest agreement.

Figure 12 compares the marginal distributions found in Figure 10. Over-estimation of ice for the low concentrations results in a different ice concentration distribution. Low ice concentrations (2/10 - 3/10) are under-represented while 9/10, 9+/10, and 10/10 are over-represented.

## 5.1 Kappa Statistic Between Analysts and MAGIC

The kappa statistic was calculated to measure the level of agreement between analysts and MAGIC (Figure 13). Both an unweighted kappa and weighted kappa was used. The unweighted kappa penalized all disagreements equally while the weighted kappa weighted greater penalties for larger differences in ice concentration estimates (e.g. far off with respect to the diagonal). The values for kappa when weighted were higher than the unweighted kappa, indicating that the spread of ice concentration estimates with respect to the diagonal were small.

The weighted kappa value assessed for all 76 polygons and all analysts as a group was 0.53, (95% CI [0.49, 0.57]), indicating mild agreement. This measure increased to 0.62 (95% CI [0.58, 0.66]), when responses were subset to only those where analysts found MAGIC segmentation was acceptable. Most of the individual analysts, however, had weighted kappa values greater than 0.6 when responses were subset to only those where analysts found MAGIC's segmentation was acceptable. In all measures, there was only one analyst who had a negative kappa statistic, indicating disagreement with MAGIC's results. This remained negative even when the responses were subset to only the MAGIC results that the analyst found acceptable.

## 5.2 Inter-Rater Reliability Between Analysts and MAGIC

For this part of the analysis, we employed Krippendorff's alpha to measure the inter-rater agreement between all eight analysts and MAGIC. The use of Krippendorff's alpha allowed us to assess how much analyst responses differed from MAGIC (and among themselves). This was important since we used MAGIC as a reference standard to compare analyst estimates in the previous section.

The Krippendorff value $\alpha = 0.762$, 95% CI [0.743, 0.780] was determined when the MAGIC segmentation results were included with all eight analysts' estimates, giving MAGIC equal weight as if it was one of the analysts in the exercise (The left most value in Figure 14). A Krippendorff value of $\alpha = 0.779$, 95% CI [0.763, 0.795] was found for the complete dataset (all eight analysts, for all polygons) with MAGIC removed. The change of $\alpha$ from 0.76 to 0.779 corresponds to a 2.23% improvement when MAGIC's calculated concentrations were removed from the group (The left most value in Figure 18).

To determine if analysts disagreed with MAGIC or with each other more, we sequentially removed each participant (analysts and MAGIC) from the group. Krippendorff's alpha was recalculated for the remaining analysts in the group. The analyst whose

removal caused the largest increase in Krippendorff's alpha was removed. That is, the analyst whose estimates disagreed the most from the remainder of the group was removed first. This process was repeated sequentially until only the two analysts whose estimates best agreed with one another remained. The results are shown in Figure 14. Sequentially removing each analyst from the group to maximize the increase in $\alpha$ suggests an order by which analysts have greatest disagreement from the rest of the group. It also identifies which individuals could potentially benefit from additional training to ensure consistency among analysts when visually estimating ice concentrations. Figure 14 shows that one analyst disagreed with all participants the most; MAGIC had the second most/largest disagreements. This illustrates that most analysts have high agreement, which leads to inter-rater reliability in ice concentration estimates.

Finally, a Krippendorff value of $\alpha = 0.814$, 95% CI [0.799, 0.829], measuring inter-rater agreement (between analysts and MAGIC), was found when the polygons were subset to only the estimates where analysts validated the segmentation results (Figure 15). The high $\alpha$ value (compared to 0.762 and 0.779) indicates agreement between analysts and MAGIC is strongest when only validated MAGIC estimates were included. Even when estimates were subset to only validated polygons (compared to when they were not), it was found that one analyst disagreed the most with the group's estimates. MAGIC's estimates had the second highest disagreement.

## 6    Comparison of Estimates Between Analysts Only Using the Group Modal Response

For this section of the analysis, the mode, or most commonly reported sea ice concentration by analysts, was used as the standard against which all other estimates were compared. This removed the dependence on segmentation results completely, isolating the assessment to only the spread and variability of estimates between analysts.

A polygon was assigned a single modal value if there was only one mode in ice concentration estimates. If there was more than one mode, then both were valid, and spread was determined using the closest modal value. For example, if a polygon had the modes 5/10 and 6/10, then an ice concentration estimate of 4/10 was considered 1/10 under-estimation away from 5/10 and 7/10 was considered 1/10 over-estimation. In the event that there was two modes with a gap, such as 4/10 and 6/10, then the midpoint between modes was used (e.g. 5/10). For three modes, the middle modal value was used.

A contingency table was produced to compare the spread and variability of all responses against the mode (Figure 16). The contingency table shows that, even in the absence of segmentation, analysts tend to over-estimate ice concentrations compared to the modal estimate by the group for low concentrations (1/10 - 3/10). The largest spread in ice concentration estimates is found at 5/10 ice concentration. The spread of the analysts estimates away from the modal value, by analysts, for all ice concentration categories, follows a normal distribution when all responses are collapsed. Figure 17 shows the difference in the marginal distributions in Figure 16.

# 7 Accuracy of Ice Concentrations in Canadian Ice Service Ice Chart Polygons

This section focuses on the accuracy of ice concentrations in polygons from the perspective of chart users, such as the shipping industry, modelers, climatologists, and other researchers. For these interpretations, we assume that the results of this exercise extend to visual estimation of ice concentration in all Canadian Ice Service ice charts. Recall that the main research question of this study was to determine how reliable the ice concentration estimate in a polygon in an ice chart is–that is, how often is the ice concentration given in the ice chart actually that ice concentration in the SAR image used to create the chart. This is assessed by using the producer's and user's accuracies (refer to Section 4 for more detail).

We first determined the producer's accuracy in order to assess the accuracy of the ice concentration estimates in the charts. The producer's accuracy gives us the probability that an analyst will assign a polygon the correct (SAR) ice concentration category. Figure 19 shows the producer's accuracy, derived from Figure 10. The producer's accuracy is low overall. For example, the producer's accuracy for 8/10 shows that analysts correctly label a given polygon as 8/10 (according to MAGIC) 39% of the time. Analysts have a rate of accuracy of 38%, 95% CI [33%, 43%], overall in estimating ice concentration to the exact tenth; this increases to 84%, 95% CI [80%, 87%], when the condition for accuracy is relaxed to +/- one tenth.

We then evaluated the user's accuracy (Figure 20) derived from Figure 10. The user's probability gives us the probability that the ice concentration assigned to a polygon in SAR has the correct ice concentration value (as estimated by the analysts). The user's accuracy for all ice concentration categories is 0.39, 95% CI [0.34, 0.43]. This increases to 0.84, 95% CI [0.80, 0.87], when the accuracy is relaxed by +/- one ice concentration category. The lowest value is for 9/10 ice concentration, which is 0.77, 95% CI [0.65, 0.85]. Both the user and producer's accuracies increase when the exact category matching is relaxed. (Exact values of the user's accuracy can be found in Table A3 and Table A4).

Most ice concentration categories have similar producer's accuracy scores but varying user's accuracy scores. The large confidence interval range is due to the small sample sizes. Sample sizes ranged between $n = 29$ to $n = 78$ for the ice concentration categories shown. A tighter confidence interval would require a greater number of polygons in this study or more analyst participation. The size of the confidence interval range is not due to the variability of estimates by analysts.

Figure 20 also shows that the over-estimation of ice concentration by analysts results in higher accuracy for lower ice concentration categories. That is, since analysts tend to over-estimate ice concentration, the hits (and hit rate) for this overestimated ice categories are also increased. Low concentrations, on the other hand, are more likely to be accurately estimated by analysts.

# 8 Discussion and Conclusions

In this study we analyzed the distribution of ice concentrations visually estimated by analysts and forecasters using SAR imagery in the HH polarization. Visually estimated ice concentrations were compared against three different standards: automatically calculated ice concentrations, automatically calculated ice concentrations that were validated by analysts, and the mode of visually estimated ice concentrations. In all three cases, visually estimated ice concentrations were over-estimated for low ice concentration categories (1/10 to 3/10) and had high variability for middle ice concentration categories (5/10, 6/10). In general, the ice concentrations estimates were consistent within analysts, and the analysts estimates were overall in agreement

with the automated segmentation estimates (as shown by Figures 7 and 10, and the high values of the kappa statistics in Figure 13).

The analysts' ice concentration estimates compared to the automated segmentation estimates exhibit an over-estimation (for all ice concentration categories evaluated). This result was achieved not only when considering all polygons, but also when considering solely the polygons for which the automated segmentation was validated by the analysts (compare Figures 7 and 10). Although these results suggest that analysts routinely over-estimate ice concentration values, it is also worth noting that the segmentation algorithm, when presented with clear images, consistently under-estimated the ice concentration by classifying ice pixels as water. This may have practical implications for automated segmentation of remotely sensed images for deriving ice concentration values.

When analyzing consistency between analysts, it was found that seven out of eight analysts strongly agreed with one another on the ice concentration estimates; one of the analysts was found in strong disagreement with the others, and the automated segmentation algorithm ranked second, in terms of disagreement (Figure 14). This was also the case when subsetting the polygons to only those that analysts determined were valid (Figure 15).

Despite the small spread between analysts estimates, it was found that ice concentration estimates from the individual analysts can vary by as much as four ice concentration categories away from the modal value. Moreover, low to intermediate ice concentrations (2/10, 3/10, 4/10) were slightly overestimated by the analysts (Figure 16) when compared to the modal value.

Finally, the accuracy of the analysts ice concentration estimates against SAR images was assessed by the producer's and user's accuracies. The probability that a given polygon in an ice chart was assigned the correct (SAR) ice concentration is 39%, 95% CI [34%, 43%]. This increases to 84%, 95% CI [80%, 87%], probability that the true ice concentration is within 1/10th of the ice concentration assigned to the polygon. Analysts have a rate of accuracy of 38%, 95% CI [33%, 43%], overall in estimating ice concentration to the exact tenth; this increases to 84%, 95% CI [80%, 87%], when the condition for accuracy is relaxed to +/- one tenth. Many chart products by Ice Services report ice concentration in ranges, rather than specific tenths; therefore, 84% shows good analyst skill.

Analysts typically start with the most recent chart when producing a new ice chart. This is done to ensure consistency and continuity between ice charts, and prevent fluctuations and variability in how polygons are drawn, or the information that they contain due to variability in analyst interpretation of SAR. In the past, analysts have carried forward the previous day's ice concentration for a given polygon, unless the analyst estimated what they felt was a significant difference in ice concentration compared to the previous chart. Nowadays, analysts are assigned specific areas to produce charts for over time to ensure better consistency. Therefore, ice concentrations in charts may exhibit higher agreement between analysts' estimates than they do without a reference chart. In a future study, analysts could be given an ice concentration estimate and then asked if they agreed or disagreed with the estimate to see if the result changes based on what the estimate given was.

It is possible to infer some preliminary comments about the use of segmentation for automatically segmenting ice and classifying ice concentrations. Overall, analysts found the segmentation results were good but consistently missed strips and patches, or new ice. Perhaps this was due to the selection of images towards the summer months; sea ice floes in the summer

can be covered by surface melt and/or characterized by new ice growth. Both of these could have impacted the ability of the algorithm to capture all of the ice. This led to consistent under-estimation of sea ice concentration by the segmentation algorithm. Using both HH/HV polarizations would have yielded a better segmentation of the sea ice in the images but were not used in this study.

Another possible factor that may contribute to the accuracy of ice concentration estimates is the size of the polygons. A large polygon will require an analyst to zoom out to view a larger geographic area. The size and shape of the floes inside the polygon may also have an impact on the accuracy of ice concentration estimates. The size of sample polygons used in this study were smaller than polygons found in corresponding ice charts and daily charts (not shown), which indicates the possibility that analysts had to zoom in to the image more than they would for regular ice charting. It is unknown if the size of the polygon
affects the estimation of ice concentration within the polygon.

Another area of interest not investigated in this study was the potential for variation in how analysts define a polygon. In this study, analysts were presented with pre-defined polygons to isolate variability in responses to their ice concentration estimates only. Perhaps, if the size of a polygon impacts an ice concentration estimate, then how an analyst defines a polygon may also impact their ice concentration estimate.

During this study, interest was voiced by the CIS Operations analysts and forecasters who participated in this study on using the tool developed as a training tool for new analysts. The tool could be used to measure if, over time, analysts' variability converge towards a common value from training. There was also interest and potential in using this tool as an ISO metric; quantifying the level of agreement between individual analysts provides a reliability measure of the spread of estimates of the group.

The differences between skilled human interpretation of ice concentration and automated algorithms needs to be better understood before automated ice classification schemes can be widely adopted in operational ice services. With respect to estimating ice concentration in this study, this means understanding why the analysts disagreed with the automated output (181 times out of 608 responses; refer to Figure 6).

The variability of ice concentration estimates can impact end users as the resulting distribution of ice concentrations changes.
Figures 9, 12, and 17 indicate that low concentrations are under-represented and high concentrations are over-represented in visual estimation. This implies that conditions may be less severe than the Canadian Ice Service ice charts show. This could result in ships being prevented from going into areas that they normally would be able to enter; however, Ice Service charts are produced for maritime safety, and therefore, err on the cautious side. Climatological studies that rely on Canadian Ice Service ice charts rely on visually estimated ice concentrations (the vertical marginal distributions in Figures 7, 10, and 16).
Prior to this study, there was little understanding of the underlying distribution of 'actual' ice concentration (the horizontal marginal distributions in Figures 7, 10, and 16). The uncertainty associated with ice concentration estimates can also impact the quality of sea ice forecasts (when ice concentration analyses are used to initialize a sea ice model), and should be a factor in the interpretation of the results of validation studies. This study contributes to the quantitative assessment of uncertainties associated to ice concentration estimates.

*Code and data availability.* The custom made user-interface (java application) and the images (in jpeg format only) used to conduct this study are available.

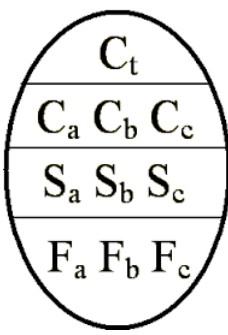

**Figure 1.** The World Meteorological Organization standard egg code used for ice charting at the Canadian Ice Service (WMO, 2014). The total concentration value, $C_t$, is the code found on the first line of the egg code. Secondary concentration values ($C_a, C_b, C_c$) can be found on the second line when more than one ice type is present.

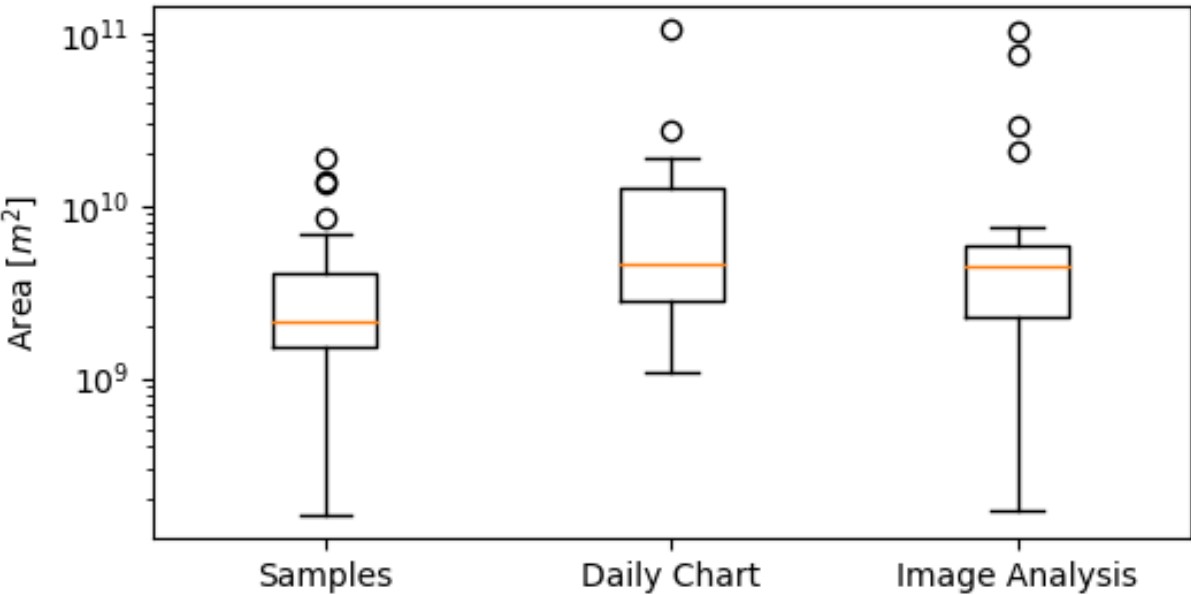

**Figure 2.** A comparison of the polygon sizes for the sample of polygons generated for this study, corresponding polygons in daily charts, and corresponding polygons in image analyses. Areas are given in $m^2$ on a log scale. The orange horizontal line indicates the median area. The upper and lower whiskers show the limits of 1.5 times the inter-quartile range. The circles indicate outlier polygon areas outside of the inter-quartile range.

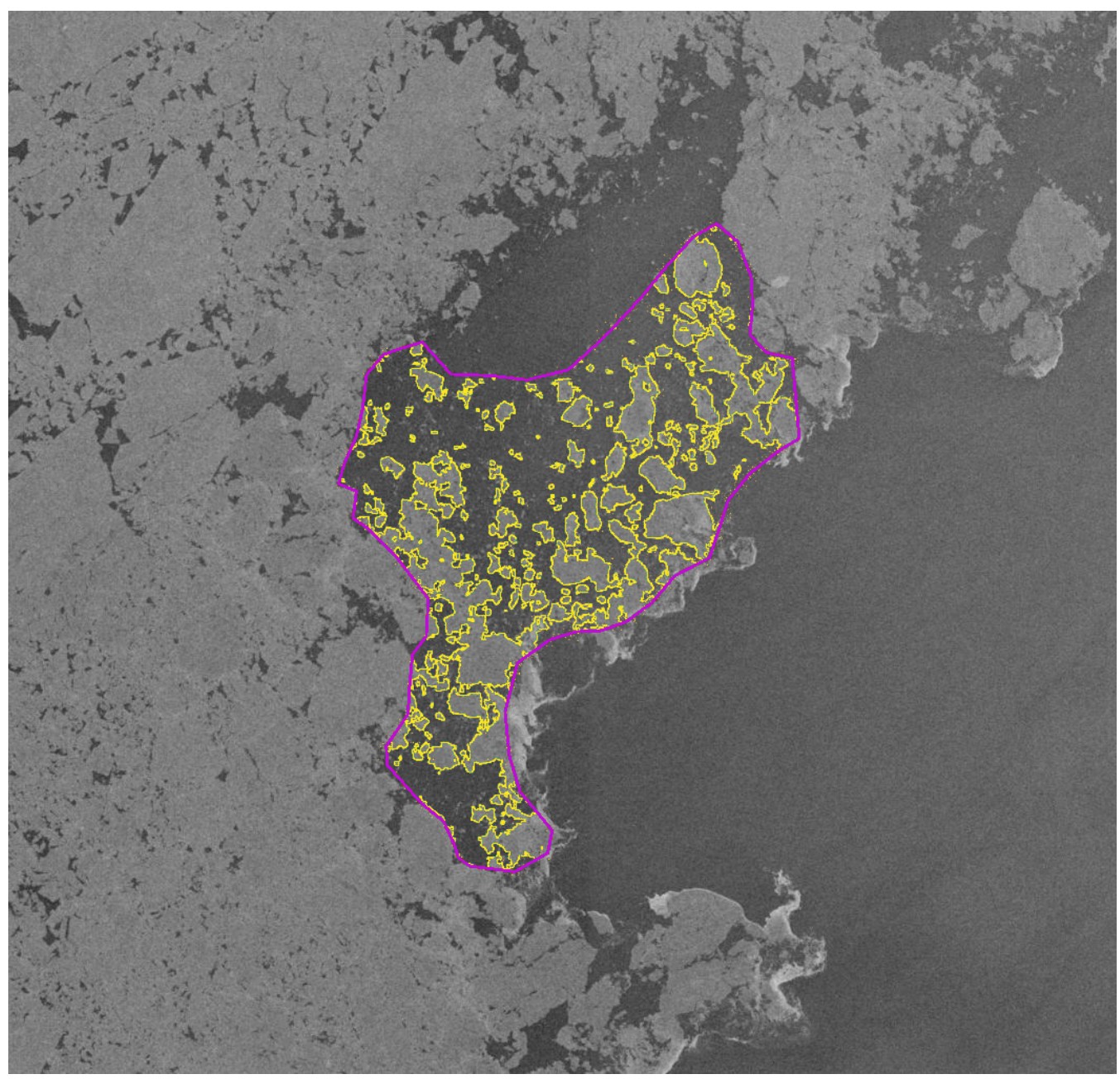

**Figure 3.** An example of a polygon in a RADARSAT-2 image (delineated area shown in magenta). The yellow outline is the calculated demarcation separating pixels tagged as ice from pixels tagged as water by the MAGIC algorithm.

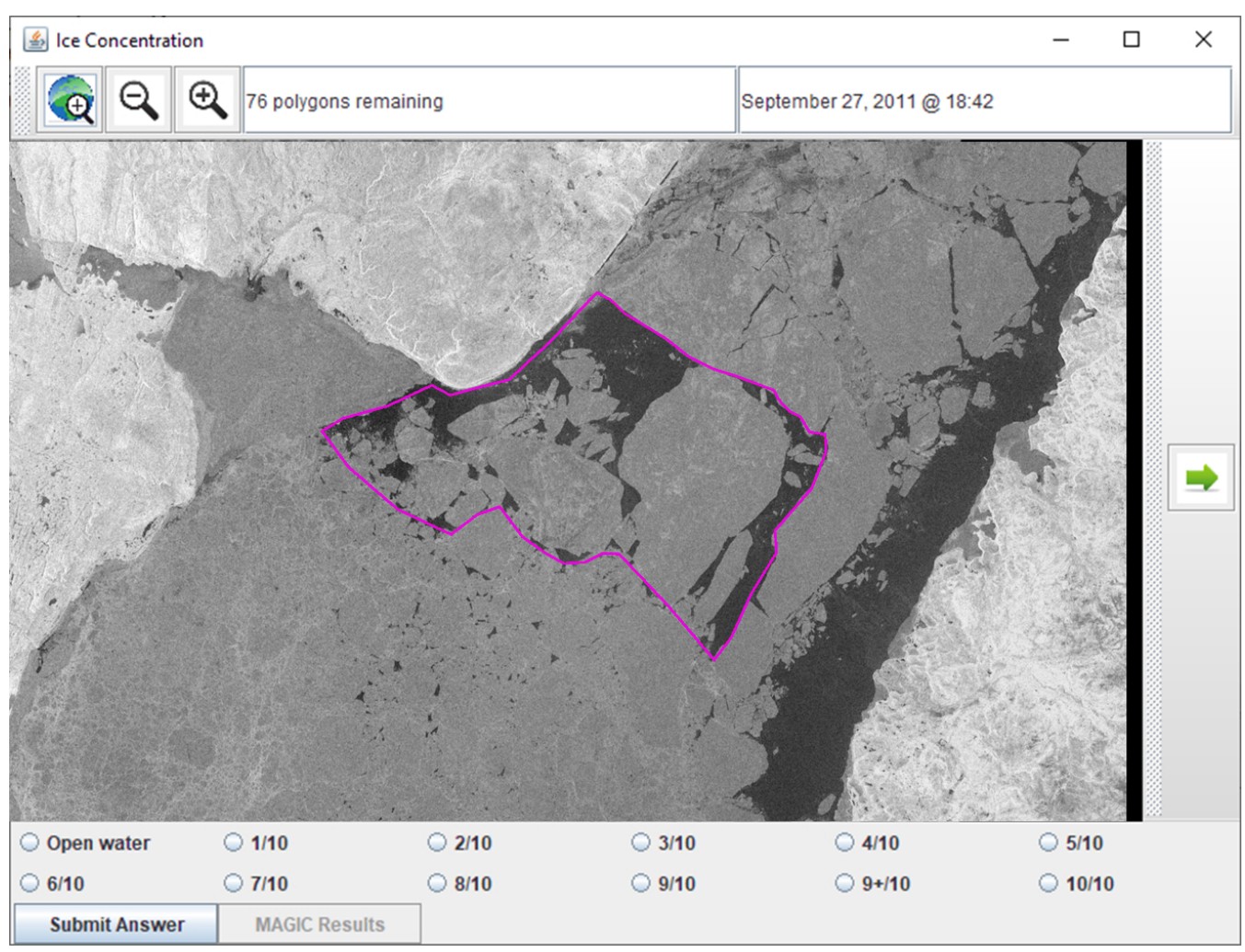

**Figure 4.** A screenshot of the user interface that the analysts were presented with. Each polygon was pre-defined so all analysts estimated ice concentrations from the same polygons. The outlined polygon was presented to the analyst, who could zoom in/out and pan the image. They were then asked to input an ice concentration value.

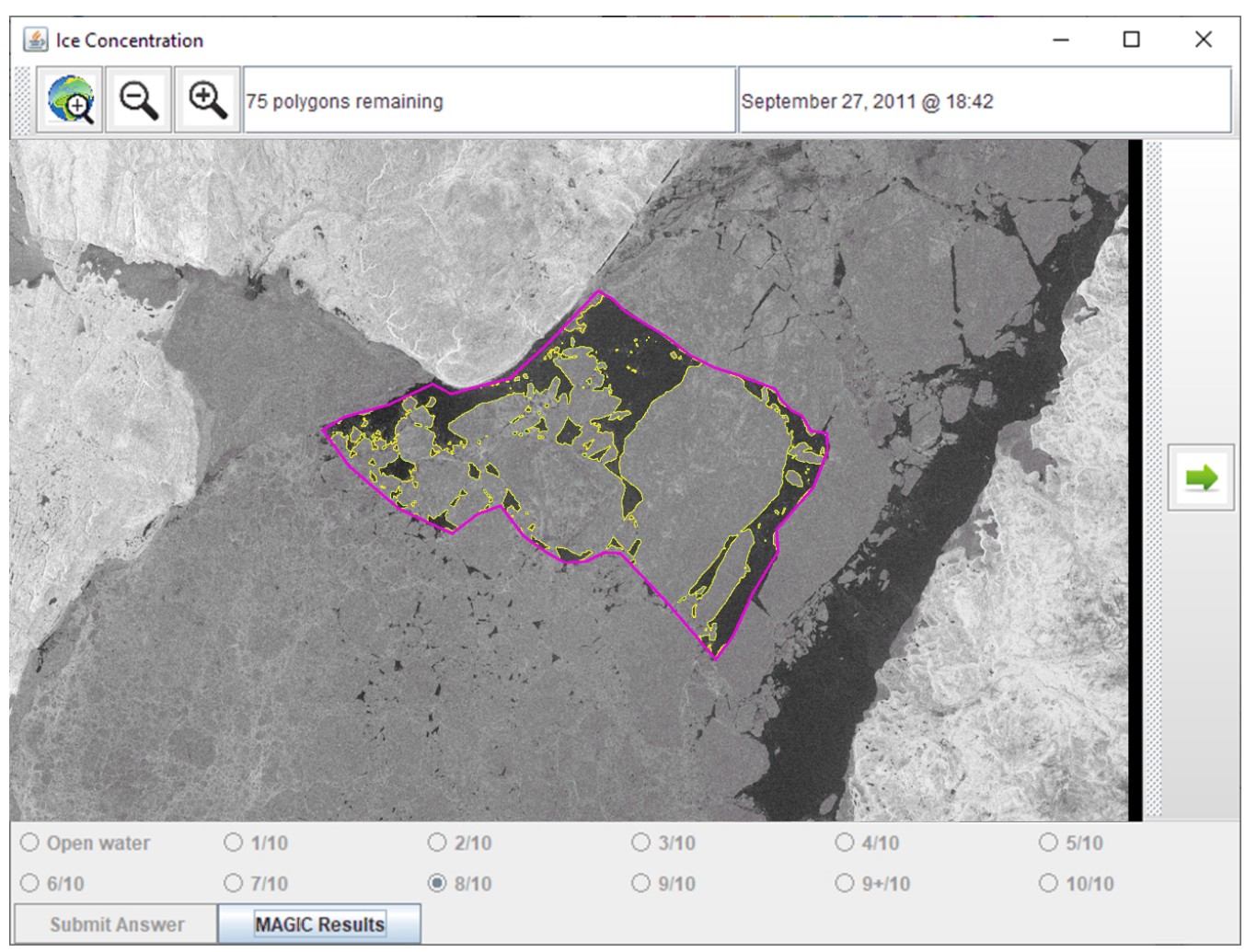

**Figure 5.** After each analyst input an ice concentration value (Figure 4), they were presented with an outline of the segmentation results from MAGIC. Analysts were able to toggle the segmentation results on/off. Answers were locked after input so that they could not be changed after the segmentation results became visible.

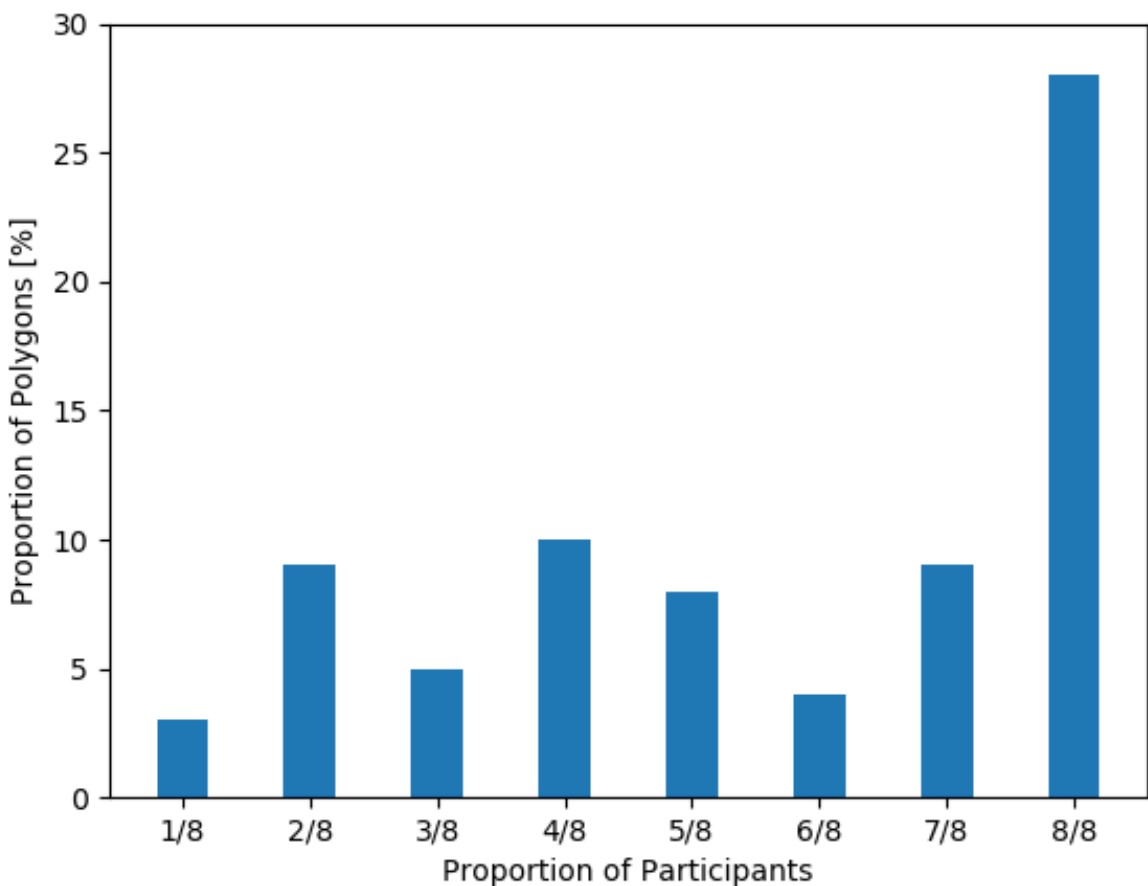

**Figure 6.** Validity of segmentation results as determined by analysts. The vertical axis shows the proportion of all polygons used in this study. The categories on the horizontal axis indicate the proportion of all (eight) participants who agreed with the segmentation results. For example, 8/8 participants reported that the segmentation results were valid over 25% of the time.

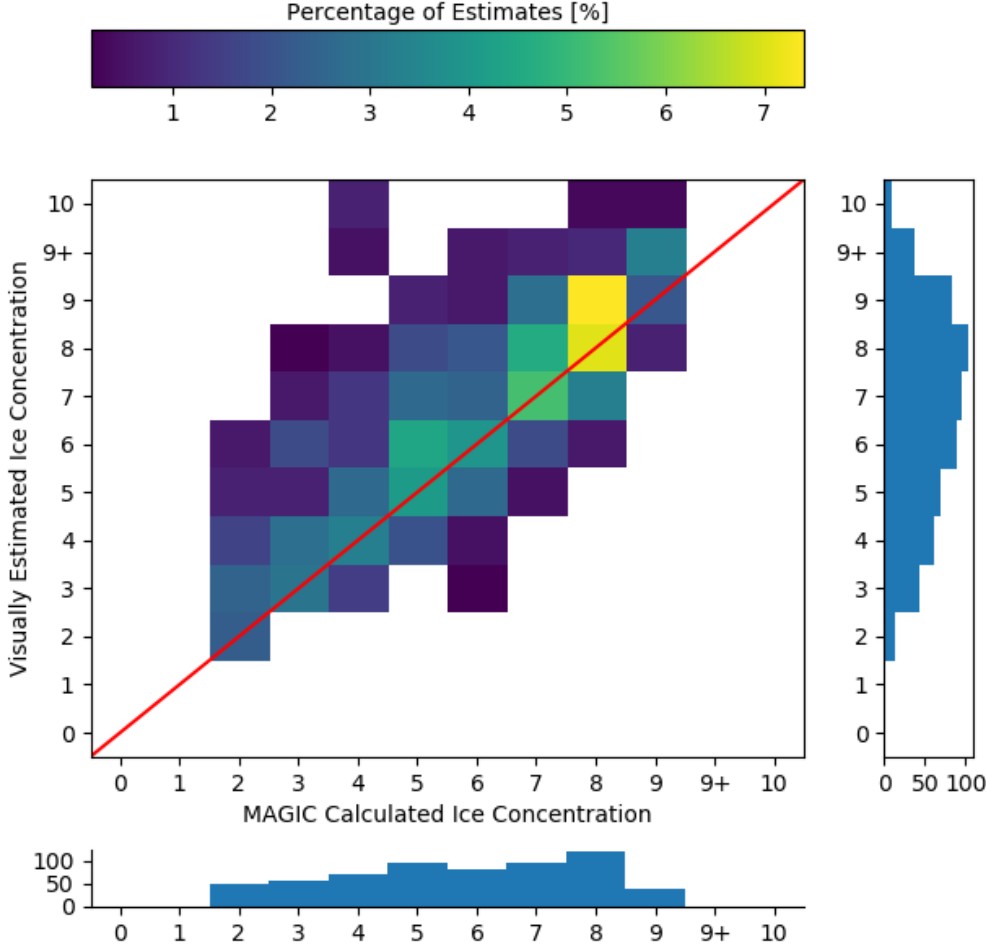

**Figure 7.** The contingency table of ice concentration estimated by the analysts (vertical axis) compared to concentration calculated from image segmentation (horizontal axis). This figure includes all responses from analysts, including responses where analysts indicated they disagreed with the segmentation results. The red diagonal line shows the location where perfect agreement between estimation and calculated concentration would occur. The histogram below shows the (marginal) distribution of ice concentrations for the segmentation by frequency (counts). There were no samples in the low (0, 1) or high (9+, 10) categories. The vertical histogram shows the (marginal) distribution of ice concentration estimates reported by the analysts by frequency. The color bar is scaled to represent the frequency of estimates as a percentage of all responses obtained.

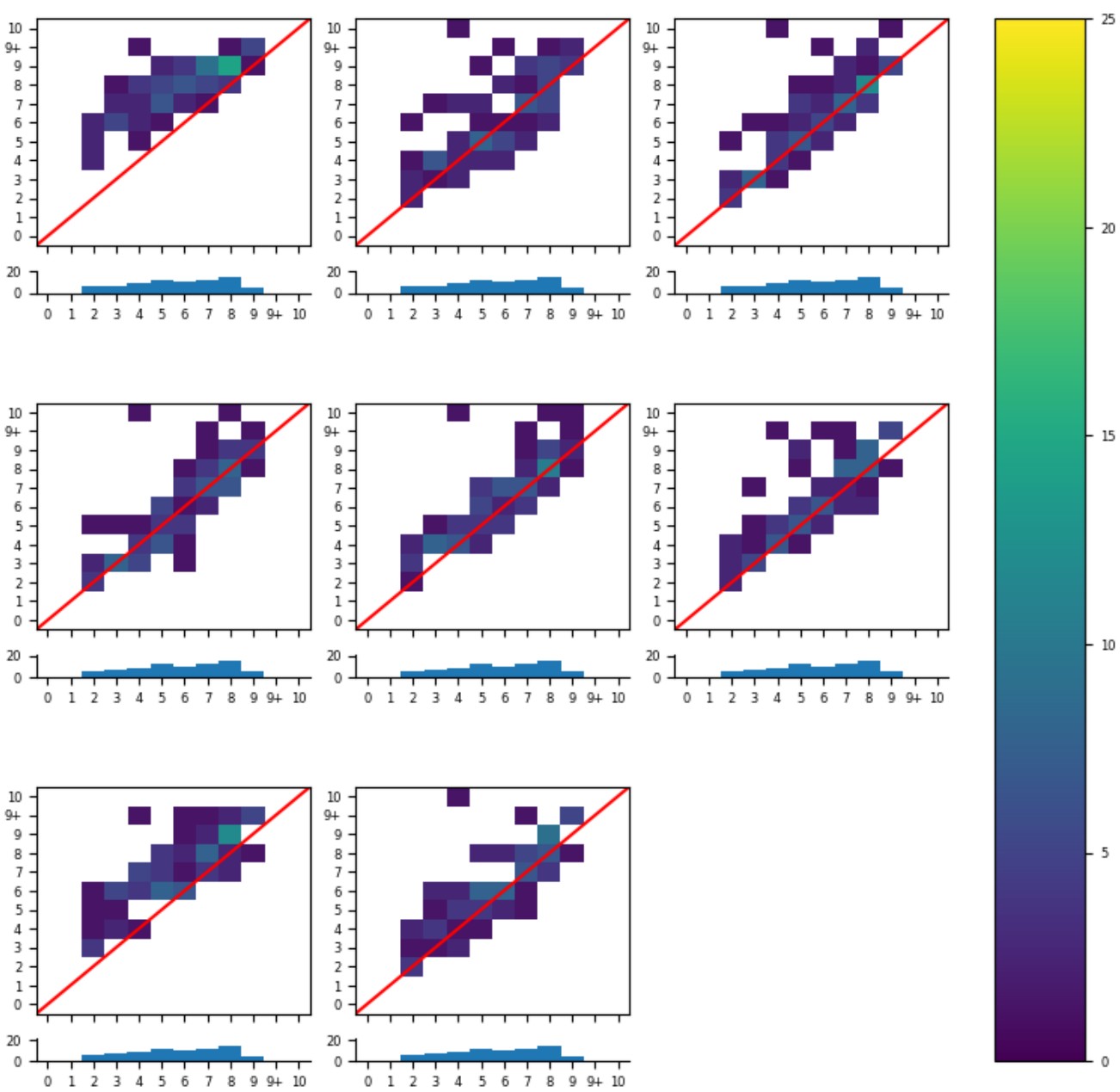

**Figure 8.** All eight analysts' individual responses. The vertical axis shows the visually estimated ice concentration by the analysts, while the horizontal shows the calculated ice concentration from image segmentation. The histogram below each contingency table shows the number of polygons in each segmentation category. The colour bar is scaled to show the percentage of all polygons in the study.

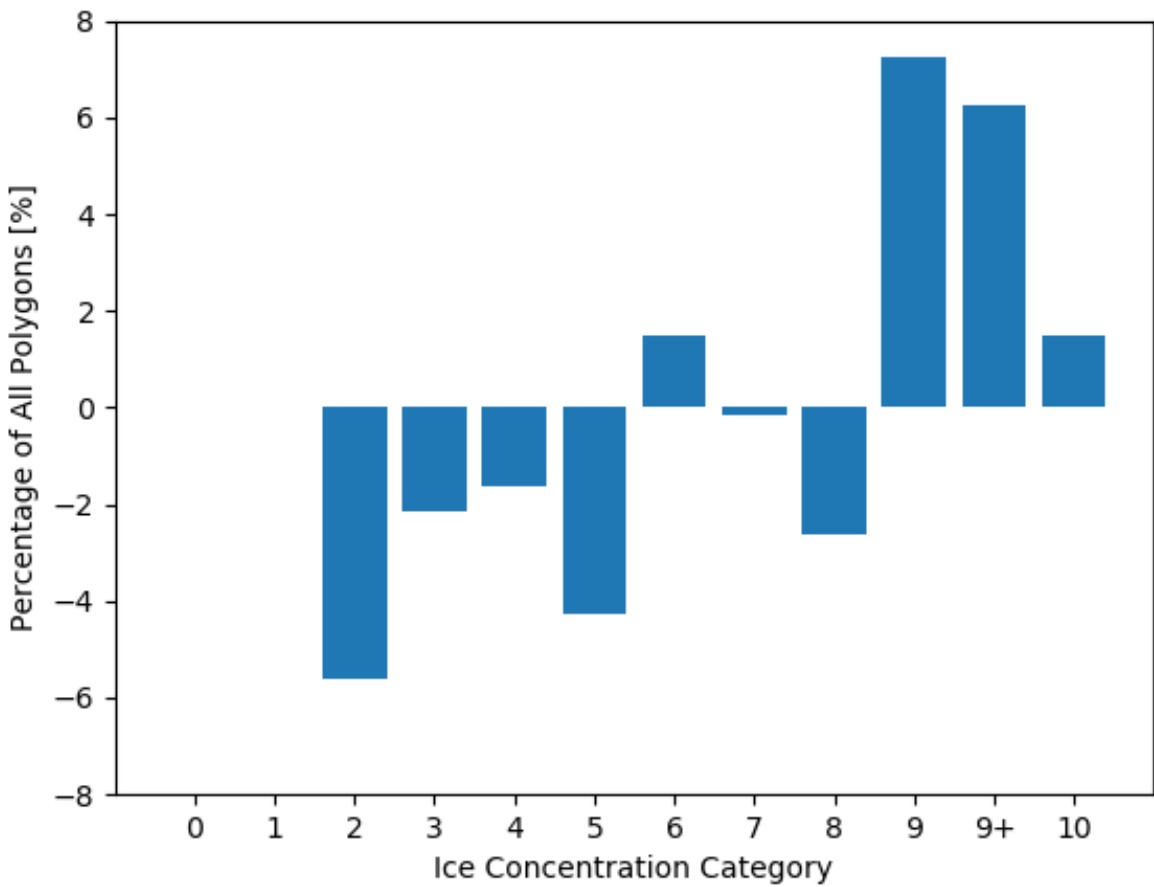

**Figure 9.** The difference in the distribution of polygons across ice concentration categories for all polygons (visually estimated ice concentration minus MAGIC). A positive (negative) value means the analyst estimated more (less) polygons than MAGIC. The vertical axis shows the difference in the proportion of polygons. The horizontal axis shows the ice concentration category.

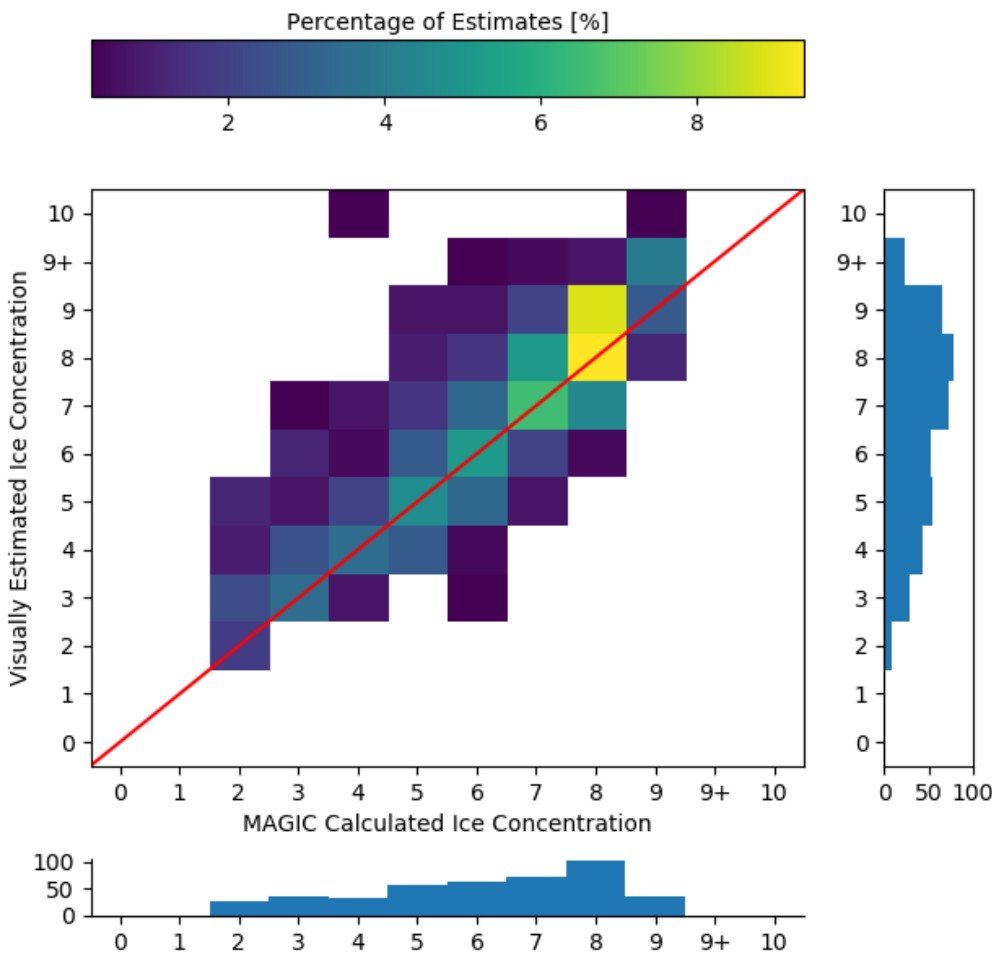

**Figure 10.** As Figure 7, but subset to only validated polygons (i.e. analysts indicated that the segmentation results from MAGIC adequately segmented the ice from open water).

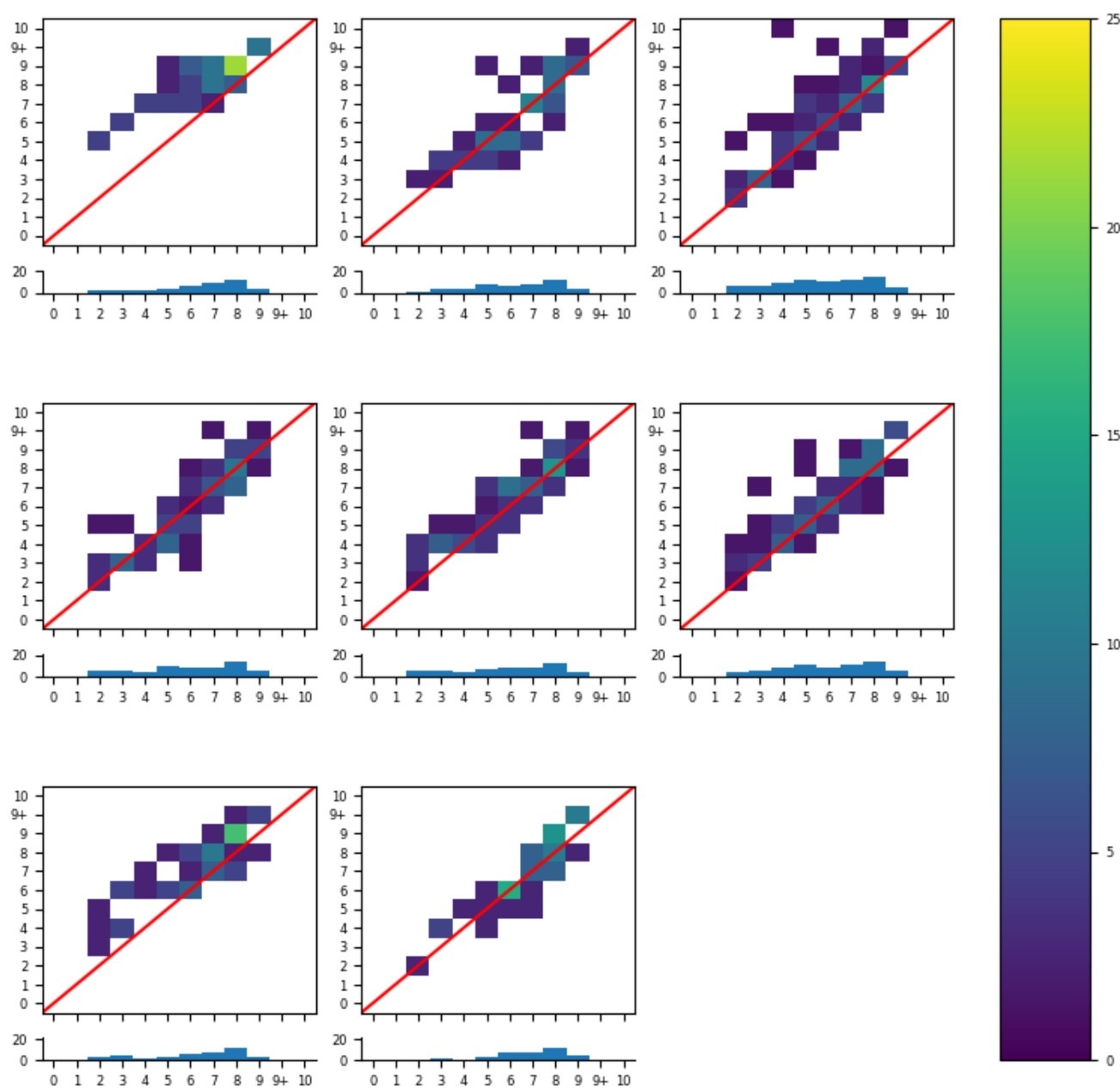

**Figure 11.** All eight analysts' individual responses, subset to only the responses where they reported the segmentation results were valid. The histogram below each contingency table shows the number of polygons in each segmentation category that they validated. The colour bar is scaled to the percentage of valid polygons.

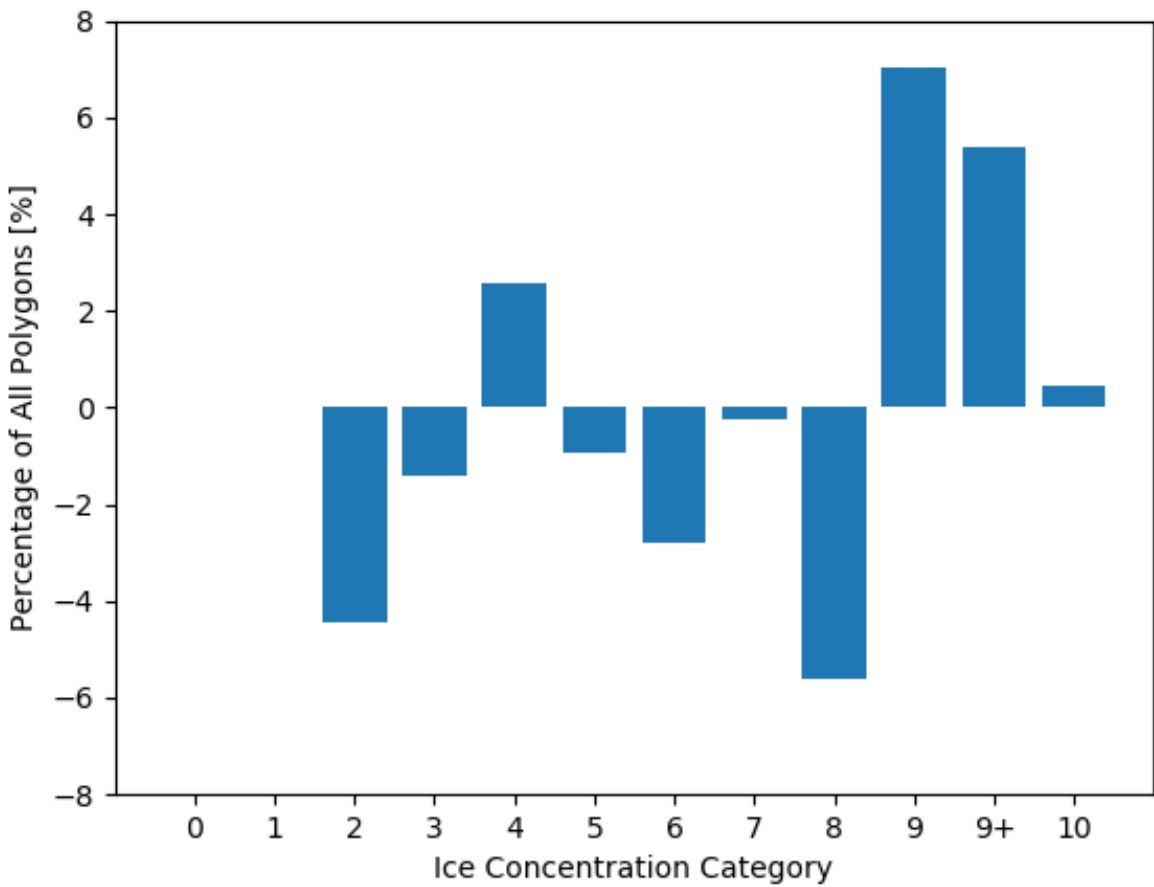

**Figure 12.** The difference in the distribution of polygons across ice concentration categories for validated polygons (visually estimated ice concentration minus MAGIC). The vertical axis shows the change in the proportion of polygons. The horizontal axis shows the ice concentration category.

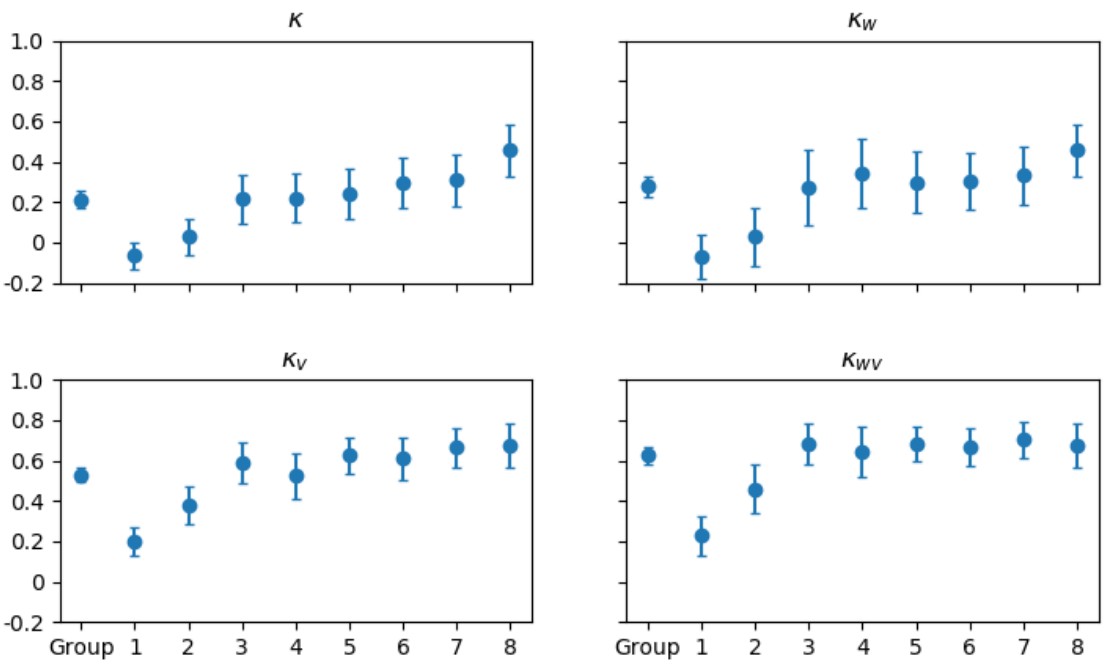

**Figure 13.** The kappa statistics of all eight analysts individually, and as a group when compared against MAGIC segmentation. The vertical axis shows the kappa value. The horizontal axis shows whether it was the group or the individual analyst. $\kappa$ includes all responses; $\kappa_v$ includes only responses where analysts determined segmentation results were acceptable; $\kappa_w$ is the weighted kappa value for all responses; $\kappa_{wv}$ is the weighted kappa for only segmentation results that were acceptable to analysts. The error bars show the upper and lower bounds for the 95% confidence interval.

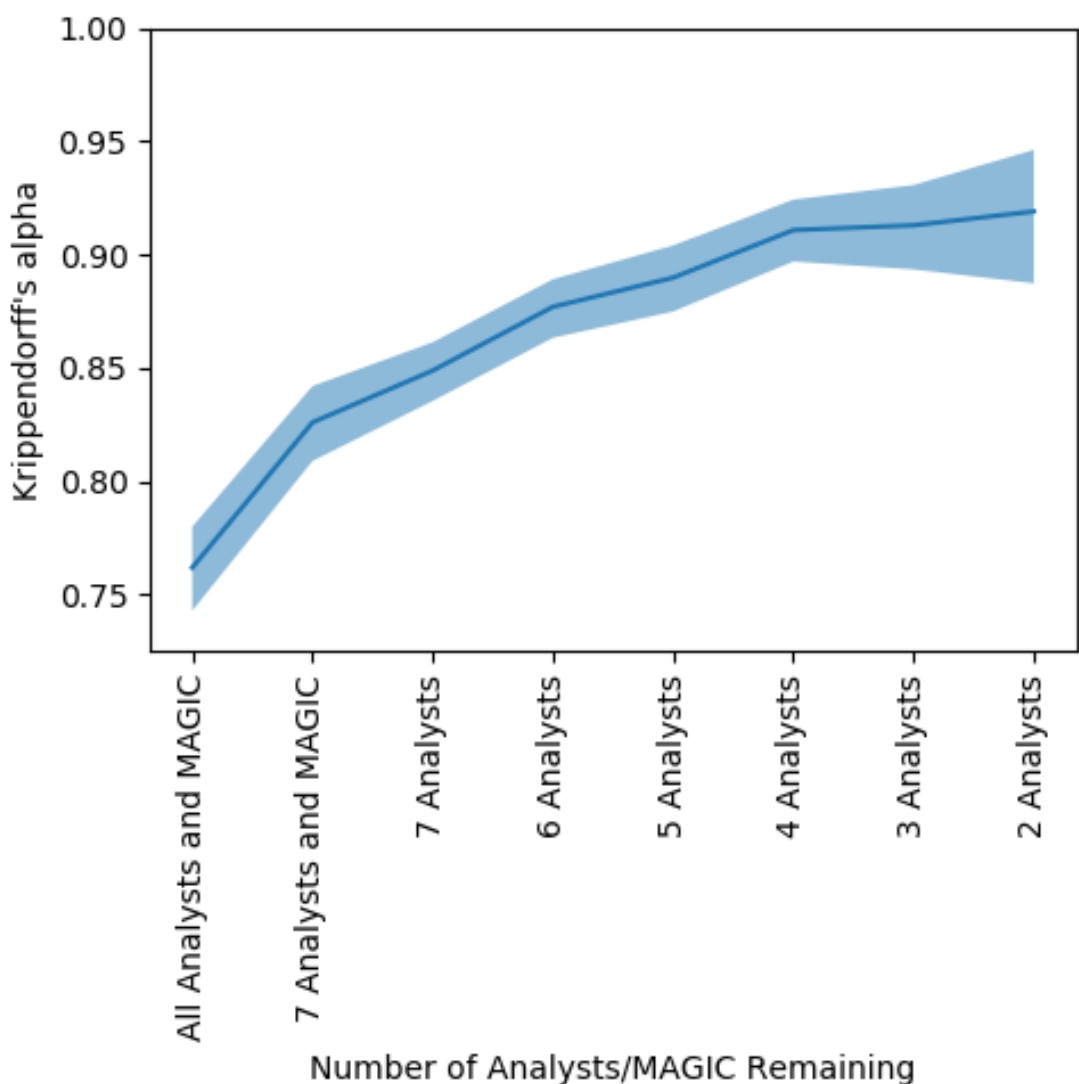

**Figure 14.** The agreement (Krippendorff's alpha) found by sequentially removing each analyst or MAGIC algorithm from the group whose absence increased alpha the most. The vertical axis shows Krippendorff's alpha. The horizontal axis shows the number of analysts/MAGIC remaining in the group considered. That is, the first entry on the left shows the Krippendorff's alpha when all eight analysts and MAGIC were included; the second entry shows Krippendorff's alpha when the analyst whose estimates disagreed the most was removed, etc., until only the two analysts whose ice concentration estimates that agreed the most with one another remained. The shaded region indicates the upper and lower bounds of the 95% confidence intervals.

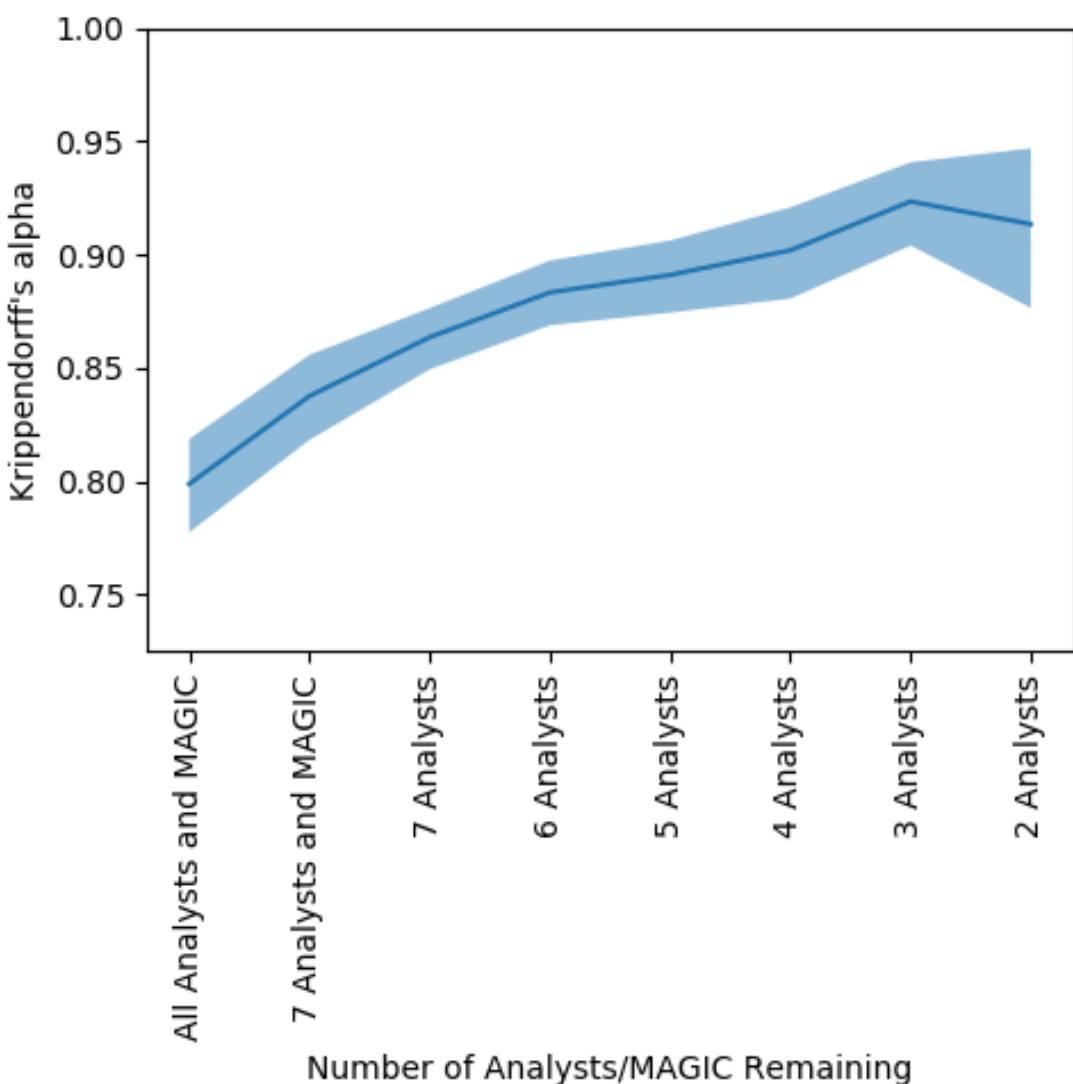

**Figure 15.** The agreement (Krippendorff's alpha) found by sequentially removing each analyst or MAGIC from the group whose absence increased alpha the most. The polygons were subset to only those validated by analysts. The vertical axis shows Krippendorff's alpha. The horizontal axis shows the number of analysts remaining in the group considered. That is, the first entry on the left shows the Krippendorff's alpha when all eight analysts and MAGIC were included; the second entry shows Krippendorff's alpha when the analyst whose estimates disagreed the most was removed, etc., until only the two analysts whose ice concentration estimates that agreed the most with one another remained. The shaded region indicates the upper and lower bounds of the 95% confidence intervals.

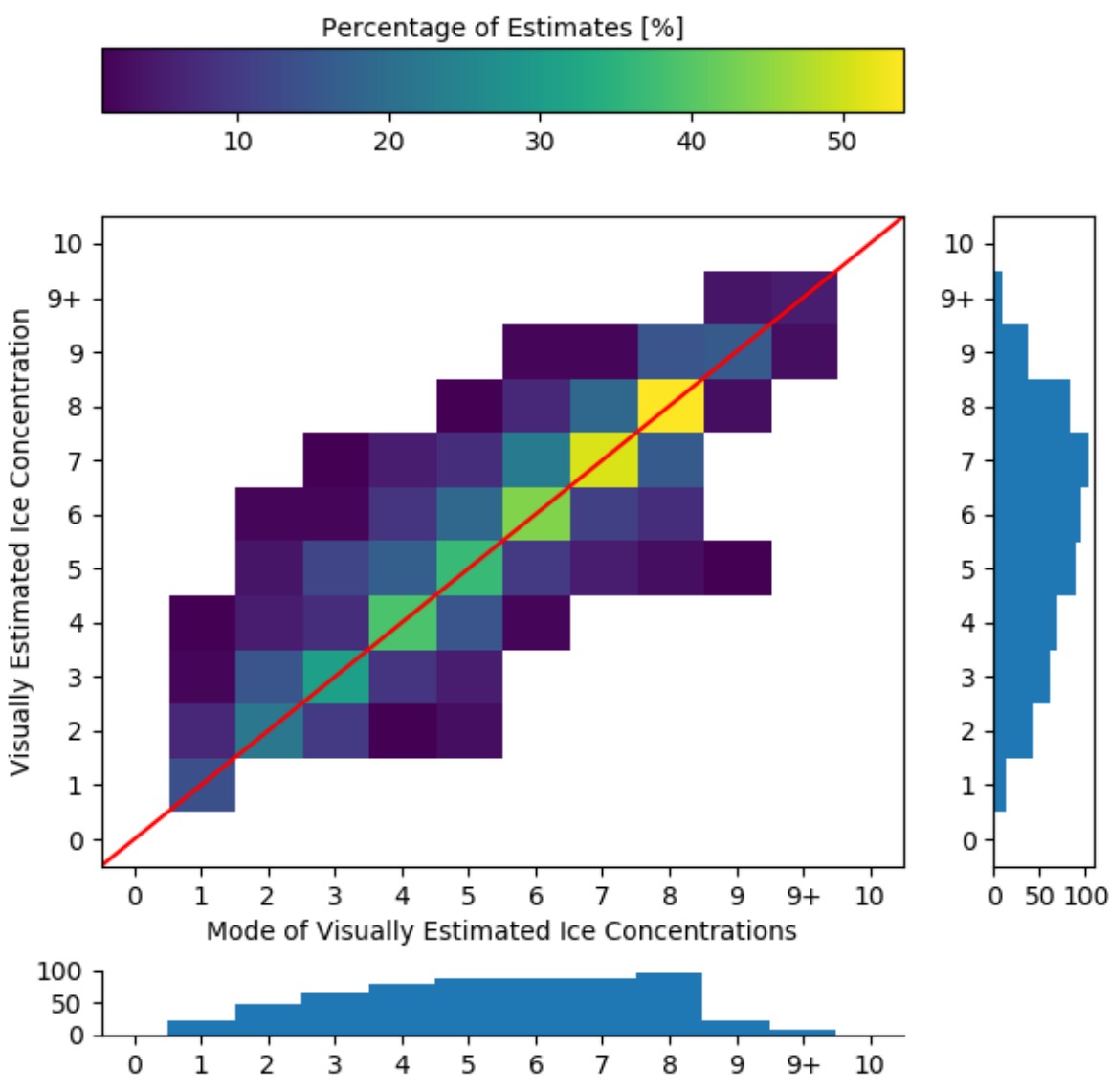

**Figure 16.** The contingency table of ice concentration visually estimated by the analysts (on the vertical axis) compared to the mode of ice concentration estimates by analysts (horizontal axis). The red diagonal line shows the location where perfect agreement between estimation and the modal estimate would occur. The histogram below shows the (marginal) distribution of ice concentration by mode. The vertical histogram shows the (marginal) distribution of estimates of ice concentration by analysts.

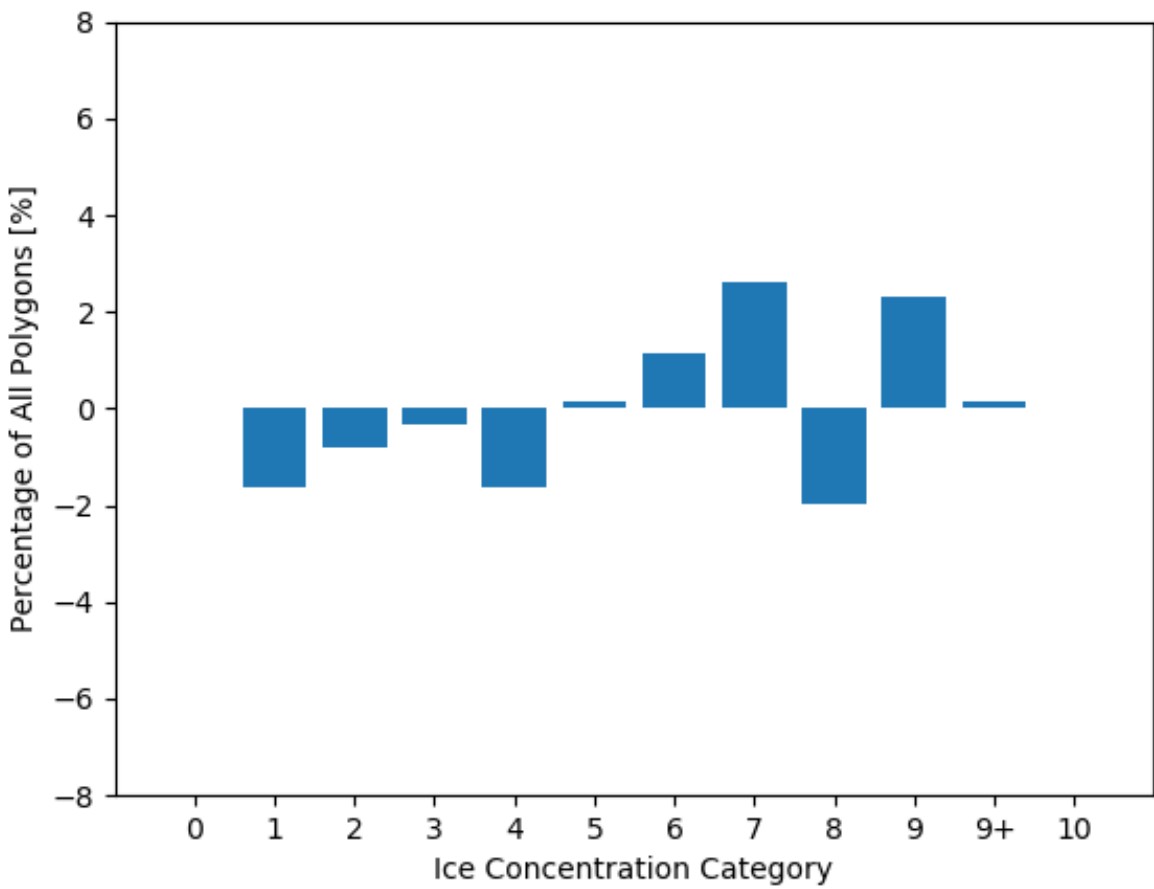

**Figure 17.** The difference in the distribution of polygons across ice concentration categories for all polygons (visually estimated ice concentration minus the modal ice concentration estimate). The vertical axis shows the change in the proportion of polygons. The horizontal axis shows the ice concentration category.

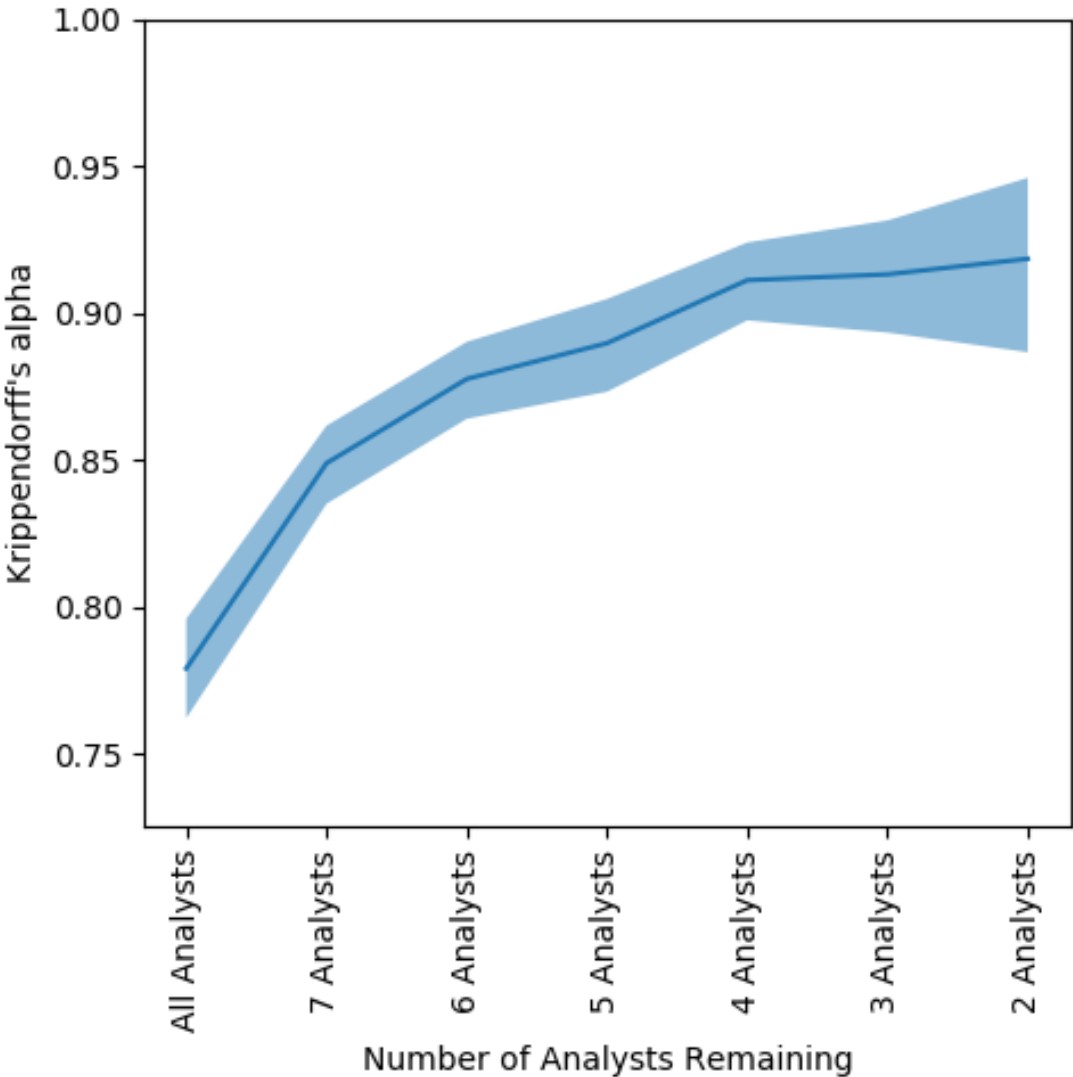

**Figure 18.** The agreement (Krippendorff's alpha) found by sequentially removing each analyst from the group whose absence increased alpha the most. The vertical axis shows Krippendorff's alpha. The horizontal axis shows the number of analysts remaining in the group considered. That is, the first entry on the left shows the Krippendorff's alpha when all eight analysts were included; the second entry shows Krippendorff's alpha when the analyst whose estimates disagreed the most was removed, etc., until only the two analysts whose ice concentration estimates that agreed the most with one another remained. The shaded region indicates the upper and lower bounds of the 95% confidence intervals.

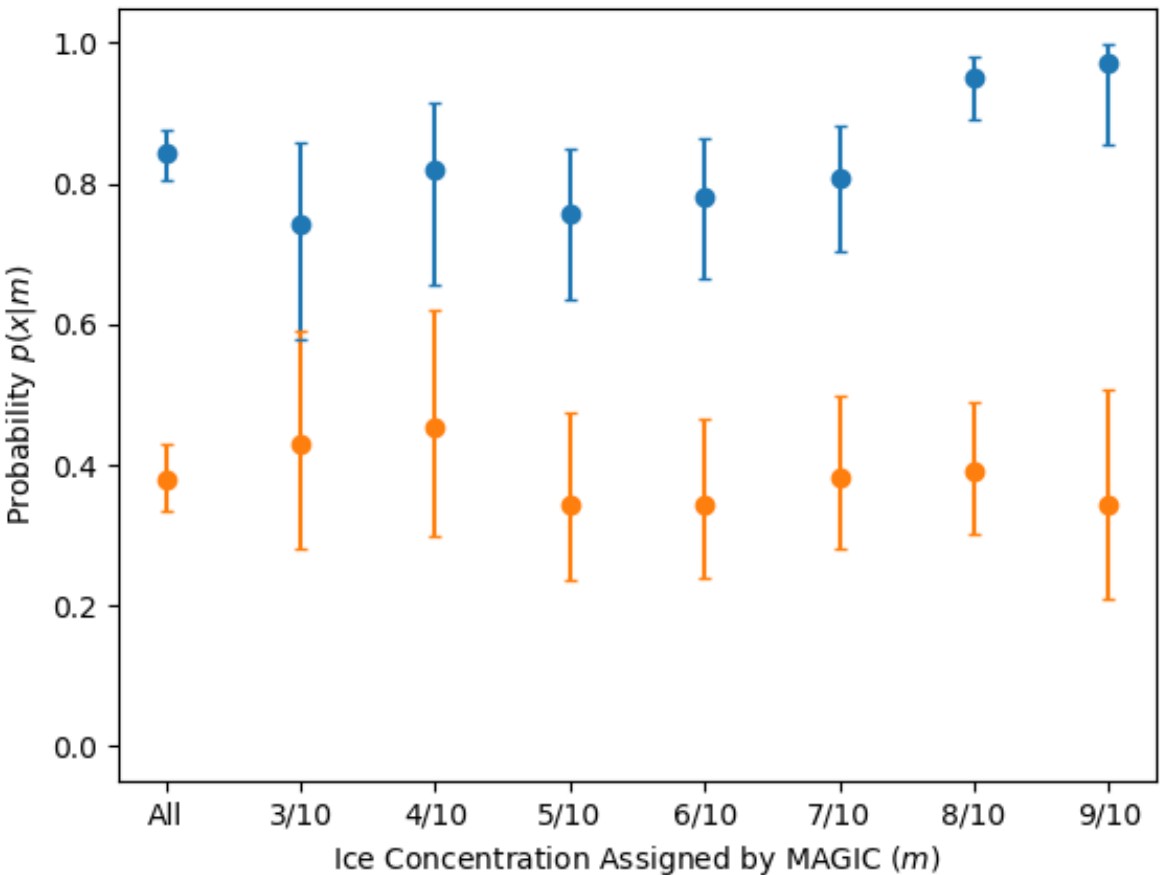

**Figure 19.** The producer's accuracy, or probability $p(x|m)$. The orange values show the probability that an analyst assigned an ice concentration value that exactly matched MAGIC, e.g. $p(x = 1|m = 1)$. The blue values show the probability that the analyst assigned an ice concentration within $\pm 1$ category of MAGIC, e.g. $p(x = 3, 4, 5|m = 4)$.

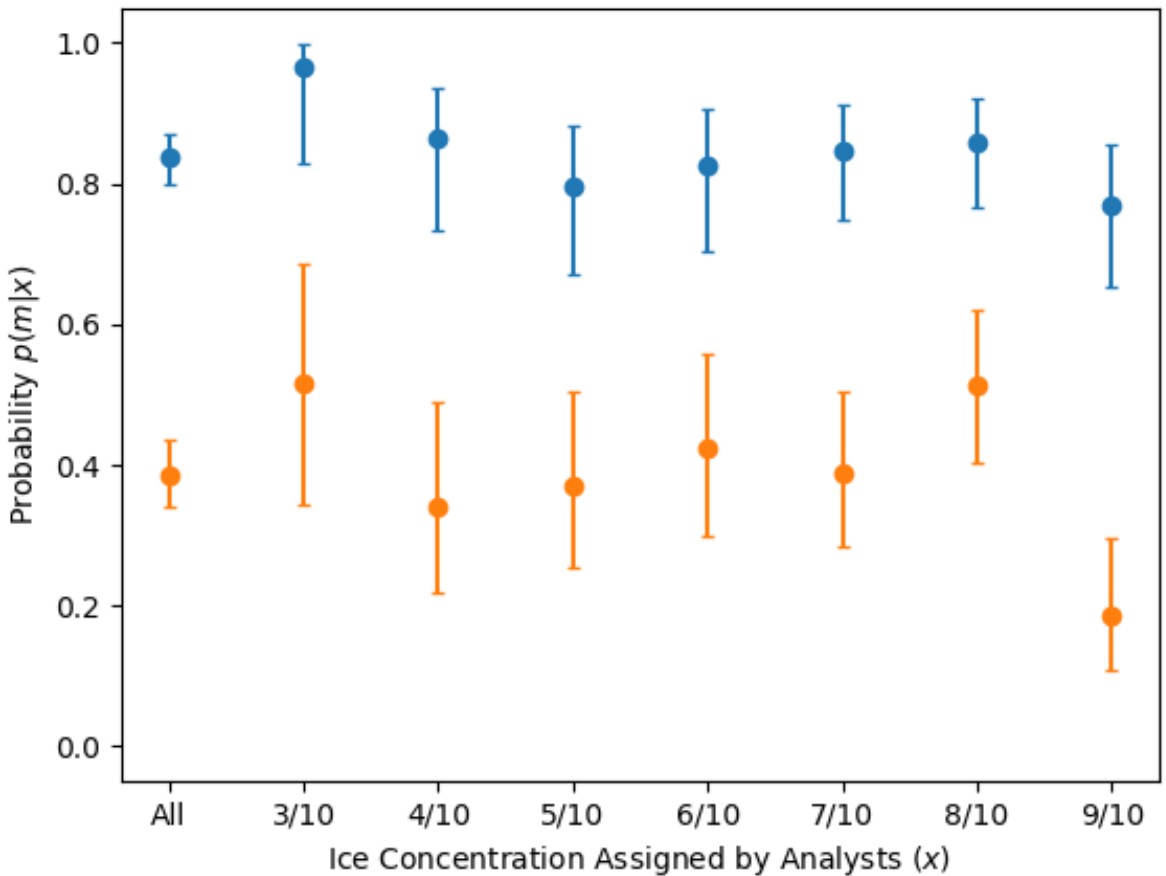

**Figure 20.** The user's accuracy, or probability $p(m|x)$ that a polygon with an assigned ice concentration will be the assigned ice concentration as calculated by MAGIC (in orange) or +/- one ice concentration category (in blue). For example, the orange values show the probability that a polygon assigned 6/10 will be 6/10 according to MAGIC; $p(m = 6|x = 6)$. Similarly, this figure reports the probability in blue that a polygon assigned 8/10 will be 7/10, 8/10, or 9/10 according to MAGIC; $p(m = 7, 8, 9|x = 8)$. Only validated polygons were used to estimate these probabilities. The confidence interval was calculated using the Wilson method for binomial probability confidence intervals and show that the true probability is found within this range at the 95% confidence level. Low ice concentrations (e.g. 0, 1, and 2/10) and high ice concentrations (9+ and 10/10) were omitted due to lack of sufficient sample sizes.

**Table A1.** Table of satellite images used in this study. The first column is the acquisition file name, and the second column contains the date that the image was acquired.

| File Name | Acquisition Date (yyyymmdd) |
|---|---|
| RS2_OK7600_PK92034_DK90462_SCWA_20090918_143522_H | 20090918 |
| RS2_OK7716_PK93164_DK91254_SCWA_20090921_144734_H | 20090921 |
| RS2_OK20363_PK220375_DK203544_SCWA_20110627_11474 | 20110627 |
| RS2_OK20392_PK226866_DK211170_SCWA_20110717_15212 | 20110717 |
| RS2_OK20404_PK230336_DK214802_SCWA_20110730_02082 | 20110730 |
| RS2_OK22206_PK237729_DK217291_SCWA_20110812_02274 | 20110812 |
| RS2_OK23037_PK240163_DK222438_SCWA_20110909_00321 | 20110909 |
| RS2_OK23041_PK240440_DK222602_SCWA_20110912_10532 | 20110912 |
| RS2_OK23045_PK241771_DK223982_SCWA_20110916_15424 | 20110916 |
| RS2_OK23046_PK242133_DK224386_SCWA_20110917_15132 | 20110917 |
| RS2_OK23049_PK243426_DK225185_SCWA_20110920_15251 | 20110920 |
| RS2_OK23057_PK247940_DK226866_SCWA_20110927_18421 | 20110927 |
| RS2_OK30832_PK303976_DK272113_SCWA_20120701_23280 | 20120701 |
| RS2_OK30832_PK303977_DK272114_SCWA_20120701_23292 | 20120701 |
| RS2_OK30832_PK303995_DK272128_SCWA_20120701_23255 | 20120701 |
| RS2_OK31008_PK307984_DK274646_SCWA_20120703_10541 | 20120703 |
| RS2_OK31008_PK308195_DK272656_SCWA_20120703_14154 | 20120703 |
| RS2_OK31009_PK305002_DK272674_SCWA_20120704_15264 | 20120704 |
| RS2_OK31010_PK305028_DK272683_SCWA_20120705_11360 | 20120705 |
| RS2_OK31011_PK305051_DK272706_SCWA_20120706_16083 | 20120706 |
| RS2_OK31011_PK305058_DK272713_SCWA_20120706_22422 | 20120706 |
| RS2_OK31011_PK308646_DK275373_SCWA_20120706_17495 | 20120706 |
| RS2_OK31011_PK308647_DK275374_SCWA_20120706_17510 | 20120706 |
| RS2_OK31013_PK306511_DK274020_SCWA_20120707_15392 | 20120707 |
| RS2_OK31013_PK305083_DK272738_SCWA_20120708_01342 | 20120708 |
| RS2_OK31017_PK305158_DK272807_SCWA_20120709_12583 | 20120709 |
| RS2_OK31017_PK309029_DK272802_SCWA_20120709_11192 | 20120709 |
| RS2_OK31145_PK310264_DK273678_SCWA_20120714_17153 | 20120714 |
| RS2_OK31145_PK306129_DK273690_SCWA_20120715_03110 | 20120715 |
| RS2_OK31146_PK306136_DK273717_SCWA_20120715_15045 | 20120715 |
| RS2_OK31145_PK306108_DK273692_SCWA_20120715_04503 | 20120715 |
| RS2_OK31147_PK306190_DK273771_SCWA_20120717_03531 | 20120717 |
| RS2_OK31362_PK312813_DK275704_SCWA_20120720_04054 | 20120720 |
| RS2_OK31364_PK308899_DK275868_SCWA_20120722_04473 | 20120722 |

**Table A2.** Continued table of satellite images used in this study. The first column is the acquisition file name, and the second column contains the date that the image was acquired.

| File Name | Acquisition Date (yyyymmdd) |
| --- | --- |
| RS2_OK31616_PK309609_DK276419_SCWA_20120731_02035 | 20120731 |
| RS2_OK32286_PK317356_DK282621_SCWA_20120815_13192 | 20120815 |
| RS2_OK32286_PK317357_DK282622_SCWA_20120815_13200 | 20120815 |
| RS2_OK32292_PK321440_DK285528_SCWA_20120820_04031 | 20120820 |
| RS2_OK33077_PK324702_DK289437_SCWA_20120917_00240 | 20120917 |
| RS2_OK33079_PK326361_DK291461_SCWA_20120918_13285 | 20120918 |
| RS2_OK41698_PK399195_DK351631_SCWA_20130704_16214 | 20130704 |
| RS2_OK41700_PK403698_DK355590_SCWA_20130705_15522 | 20130705 |
| RS2_OK41703_PK403904_DK351755_SCWA_20130707_16342 | 20130707 |
| RS2_OK41775_PK400013_DK352434_SCWA_20130709_15353 | 20130709 |
| RS2_OK41777_PK400262_DK352466_SCWA_20130710_16454 | 20130710 |
| RS2_OK41956_PK401184_DK354333_SCWA_20130721_13020 | 20130721 |
| RS2_OK42514_PK407529_DK358363_SCWA_20130730_23364 | 20130730 |
| RS2_OK43177_PK422661_DK373182_SCWA_20130816_23411 | 20130816 |
| RS2_OK43181_PK422966_DK373445_SCWA_20130818_17470 | 20130818 |
| RS2_OK43181_PK418292_DK369145_SCWA_20130819_03461 | 20130819 |
| RS2_OK43181_PK418309_DK369162_SCWA_20130819_00235 | 20130819 |
| RS2_OK43183_PK418327_DK369177_SCWA_20130819_13564 | 20130819 |
| RS2_OK43183_PK418346_DK369196_SCWA_20130819_23544 | 20130819 |
| RS2_OK43407_PK419156_DK369995_SCWA_20130820_13273 | 20130820 |
| RS2_OK43411_PK420183_DK370848_SCWA_20130822_17301 | 20130822 |
| RS2_OK43411_PK420188_DK370853_SCWA_20130823_00070 | 20130823 |
| RS2_OK43413_PK419305_DK370118_SCWA_20130823_13400 | 20130823 |
| RS2_OK43417_PK419391_DK370205_SCWA_20130825_14224 | 20130825 |
| RS2_OK43417_PK422985_DK373464_SCWA_20130825_17430 | 20130825 |
| RS2_OK43628_PK421988_DK372378_SCWA_20130830_23331 | 20130830 |
| RS2_OK43628_PK421963_DK372353_SCWA_20130830_13364 | 20130830 |
| RS2_OK44067_PK425718_DK376086_SCWA_20130910_13145 | 20130910 |
| RS2_OK44067_PK425719_DK376087_SCWA_20130910_13161 | 20130910 |
| RS2_OK44073_PK429167_DK376197_SCWA_20130914_01044 | 20130914 |
| RS2_OK44079_PK430547_DK377682_SCWA_20130916_15224 | 20130916 |
| RS2_OK44286_PK431119_DK381223_SCWA_20130921_02404 | 20130921 |
| RS2_OK44290_PK431600_DK378006_SCWA_20130922_15474 | 20130922 |

**Table A3.** Probability of a polygon being assigned correctly within $\pm 0, 1, 2, 3$ ice concentration categories (user's accuracy, or $p(m|x)$). $\hat{p}$ refers to the point estimate of the likelihood found in our sample data; lower and upper refer to the lower and upper 95% confidence intervals.

| Ice Concentration | $\pm 0$ | | | $\pm 1$ | | | $\pm 2$ | | | $\pm 3$ | | |
|---|---|---|---|---|---|---|---|---|---|---|---|---|
| | $\hat{p}$ | Lower | Upper | $\hat{p}$ | Lower | Upper | $\hat{p}$ | Lower | Upper | $\hat{p}$ | Lower | Upper |
| All | 0.386 | 0.339 | 0.435 | 0.838 | 0.798 | 0.871 | 0.937 | 0.908 | 0.957 | 0.990 | 0.974 | 0.996 |
| 3/10 | 0.517 | 0.344 | 0.686 | 0.966 | 0.828 | 0.998 | 0.966 | 0.828 | 0.998 | 1.000 | 0.883 | 1.000 |
| 4/10 | 0.341 | 0.219 | 0.489 | 0.864 | 0.733 | 0.836 | 1.000 | 0.920 | 1.000 | 1.000 | 0.920 | 1.000 |
| 5/10 | 0.370 | 0.254 | 0.504 | 0.796 | 0.671 | 0.882 | 0.907 | 0.801 | 0.960 | 1.000 | 0.934 | 1.000 |
| 6/10 | 0.423 | 0.299 | 0.558 | 0.827 | 0.703 | 0.906 | 0.904 | 0.794 | 0.958 | 1.000 | 0.931 | 1.000 |
| 7/10 | 0.389 | 0.285 | 0.504 | 0.847 | 0.747 | 0.912 | 0.944 | 0.866 | 0.978 | 0.986 | 0.925 | 0.999 |
| 8/10 | 0.513 | 0.404 | 0.621 | 0.859 | 0.765 | 0.919 | 0.949 | 0.875 | 0.980 | 1.000 | 0.953 | 1.000 |
| 9/10 | 0.185 | 0.109 | 0.296 | 0.769 | 0.654 | 0.855 | 0.908 | 0.813 | 0.957 | 0.954 | 0.873 | 0.954 |

**Table A4.** Probability of a polygon being assigned correctly for each ice concentration category (user's accuracy, or $p(m|x)$). $\hat{p}$ refers to the point estimate of the likelihood found in our sample data; lower and upper refer to the lower and upper 95% confidence intervals, calculated using a Sison-Glaz multinomial confidence interval method. For any ice concentration category where the $\hat{p}$ value for each row does not sum to 1.0, the remainder was found outside of $\pm$ three ice concentration categories from the category denoted in the first column.

| Ice Concentration | -3 | | | -2 | | | -1 | | | 0 | | | +1 | | | +2 | | | +3 | | |
|---|---|---|---|---|---|---|---|---|---|---|---|---|---|---|---|---|---|---|---|---|---|
| | $\hat{p}$ | lower | upper | $\hat{p}$ | lower | upper | $\hat{p}$ | lower | upper | $\hat{p}$ | lower | upper | $\hat{p}$ | lower | upper | $\hat{p}$ | lower | upper | $\hat{p}$ | lower | upper |
| 3/10 | - | - | - | - | - | - | 0.345 | 0.172 | 0.525 | 0.517 | 0.345 | 0.698 | 0.103 | 0.000 | 0.284 | 0.000 | 0.000 | 0.181 | 0.034 | 0.000 | 0.215 |
| 4/10 | - | - | - | 0.091 | 0.000 | 0.254 | 0.250 | 0.114 | 0.413 | 0.341 | 0.205 | 0.504 | 0.273 | 0.136 | 0.436 | 0.045 | 0.000 | 0.209 | - | - | - |
| 5/10 | 0.093 | 0.000 | 0.227 | 0.056 | 0.000 | 0.190 | 0.167 | 0.037 | 0.301 | 0.370 | 0.241 | 0.504 | 0.259 | 0.130 | 0.393 | 0.056 | 0.000 | 0.190 | - | - | - |
| 6/10 | 0.096 | 0.000 | 0.248 | 0.038 | 0.000 | 0.191 | 0.231 | 0.0115 | 0.383 | 0.423 | 0.308 | 0.575 | 0.173 | 0.058 | 0.325 | 0.038 | 0.000 | 0.191 | - | - | - |
| 7/10 | 0.042 | 0.000 | 0.166 | 0.097 | 0.000 | 0.221 | 0.194 | 0.083 | 0.318 | 0.389 | 0.278 | 0.513 | 0.264 | 0.153 | 0.388 | - | - | - | - | - | - |
| 8/10 | 0.051 | 0.000 | 0.171 | 0.090 | 0.000 | 0.209 | 0.282 | 0.179 | 0.401 | 0.513 | 0.410 | 0.632 | 0.064 | 0.000 | 0.183 | - | - | - | - | - | - |
| 9/10 | 0.046 | 0.000 | 0.169 | 0.138 | 0.031 | 0.261 | 0.585 | 0.477 | 0.708 | 0.185 | 0.077 | 0.308 | - | - | - | - | - | - | - | - | - |

*Author contributions.* AC designed the research methodology, user interface, and led the analysts through the exercise. AC also conducted the analysis, created the figures, and wrote the first draft of the paper. BC was the main editor, reviewed the skill scores used and corresponding mathematical expressions, and ensured that the skill scores were aligned with previous forecast verification methods. AT critiqued the text and figures for readability to ensure the results would be as useful as possible for end users of Canadian Ice Service ice charts. TZ was the former operations analyst mentioned in this study who created the 76 polygons in SAR imagery. JFL reviewed the sources cited. JFL and BT revised the text (particularly from a modeling perspective), and critiqued the design of the figures used. BT modified the figure captions. All authors contributed to the interpretation of the results.

*Competing interests.* The authors declare that they have no conflict of interest.

*Acknowledgements.* The authors would like to thank the eight anonymous analysts and forecasters in the Canadian Ice Service Operations group for their participation in this exercise. The authors thank Dr. David Clausi and Dr. Linlin Xu for providing guidance and allowing the use of MAGIC for this study. We also thank Nicolas Denis, Thomas Stubbs, and Paul Greco for their assistance in generating polygons and segmentation outputs for the exercise. The authors thank Natalizia Delorenzis in her assistance determining the polygon areas. We thank Dr. Alexander Komarov and Kerri Warner for their comments and edits of this paper. Finally, we thank the three anonymous reviewers whose comments and suggestions greatly improved this paper.

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
