# Peer review of "Accuracy and Inter-Analyst Agreement of Visually Estimated Sea Ice Concentrations in Canadian Ice Service Ice Charts Using HH SAR"

_The Cryosphere, 2019_

## Referee Comment (RC1) · Anonymous Referee #1 · 16 Oct 2019

This paper assesses the accuracy of CIS products and the agreement between analysts in the generation of the CIS ice charts. This type of study is relevant to the field of sea ice mapping since CIS charts can be used as training/validation data for developing automated methods. Thus, it is important to assess the accuracy of these products and understand how they are generated, their strengths and weaknesses.

This paper presents good statistical analyses of the gathered information and is a good start to assess the accuracy of CIS charts. Nonetheless, I would recommend more caution in the discussions section since with more analysts, there could be more of them that disagree with the general consensus than just one out of eight.

[Figure]

Also, does the methodology implemented here reflect how the CIS products are generated? Since CIS offers daily products, they surely refer to previous day analysis to determine ice concentration. This would greatly impact the general agreement between analysts. This should be discussed in section 8.

Another analysis that I think should be added to this study, and the authors partially addressed it in section 8, is how the agreements vary with ice types and not only ice concentrations. Ice type is also a significant variable for shipping and weather/climatology modeling. I suggest adding details in section 3 about the distribution of ice concentrations and ice types for all the different polygon to show how this study reflects the ice conditions seen in Canadian waters.

I think in a future study, the authors should look at how MAGIC compares to CIS charts generated operationally to assess the agreement between both products with a higher statistical significance than just imagery with high contrast. The authors should elaborate a bit more in the discussion section how MAGIC performs in low contrast or different ice conditions and how this could impact results of this study.

That being said, I recommend that this paper should be published after revisions by the authors since it brings important information about the CIS charts to the science community.

Specific Comments:

P.4L.10: add "(floe size)" after "predominant form (F) of ice" to clarify P.16L.33: I imagine that 7/10 is considered 1/10 overestimation in this case? Please clarify.

―――――――――――――――――――――――――――――――

---

## Referee Comment (RC2) · Anonymous Referee #2 · 24 Oct 2019

**General Comments**

The paper presents a comparison of ice analyst assessments of sea ice concentration values at the Canadian Ice Service, both between analysts and against an automated retrieval method. It is well written and I see only some minor issues that can be fixed or explained better. The study is limited to Canadian Ice Service analysts, and finds very good agreement between them. With more analysts, including from other ice services, it is likely the range of opinions will increase. However running this type of exercise is time intensive, so only a limited number of such studies have been performed (i.e. Karvonen et al, 2015). Other studies have tried examining the areas

where ice charts overlap, for example a comparison between Norwegian and Danish ice charts at *https://www.ecmwf.int/sites/default/files/elibrary/2018/17968-satellite-observations-sea-ice-concentration-and-drift.pdf* (p.14-15). More of this type of paper evaluating ice chart data are needed to provide quantitative measures of uncertainty that can be used when ice charts are used as training data for automated classification algorithm developments and as climate records.

**Specific Comments**

P2 L2 Analysts at the CIS delineate boundaries around regions with similar ice conditions. This raises the question, what was the rationale for the example polygon areas in Figures 3 and 4 as these contain dissimilar ice conditions?

P2 L28-34. This uncertainty is not limited to ice charts, automatic satellite-derived sea ice concentration products also have varying levels of uncertainty.

P3 L31 and P5 L25. How does limiting the analysts to only HH affect their choice of concentration? As noted at L29, the addition of HV has improved distinguishing between ice and water, and Canadian analysts have had access to this from RADARSAT-2 for over a decade. One would imagine limiting to HH-only would be generating additional analyst uncertainty. It would be interesting in a further study to provide one set of images as HH, and another as HV.

P4 L8. The egg code is a World Meteorological Organization international standard. Reference should be made to WMO publication 259, Sea Ice Nomenclature.

P6 Figure 3. The example provided shows an area containing both dark areas of open water and lighter areas of concentrated sea ice. The open water areas are large and contiguous enough that most ice analysts would delineate them as separate areas, and some explanation is needed as to why these should be grouped with large areas of concentrated ice. MAGIC looks to have done a good job of marking these open water areas.

[Figure]

P8 Figure 4. Similar question, why does the polygon contain areas of areas of very open drift ice that are continuous with adjacent areas outside the polygon?

P13 L11 - P14 L3. 181 disagreements, or almost a third, is a large proportion. MAGIC provided lower estimates in 175 of cases, but analysts tend to err on the conservative side, and if the mix of ice concentrations shown in the example image polygons is representative of the dataset, it is to be expected that there would be more disagreement, as shown by only 36.8% of examples having unanimous agreement between analysts.

P18 L31-32. A 84% at plus/minus one-tenth accuracy shows very good analyst skill, most ice services only work in concentration ranges.

P19 L22-23. Agree, the definition of the polygon is critical. The examples provided appear somewhat ambiguous so some explanation as to why the areas shown in the examples are as they are would be useful.

P19 L35, P20 L1. The ice charts are for maritime safety so will err on the cautious side.

**Technical Corrections**

Figure 2. Difficult to view at this size and could be larger.

Figures 2 and 4. The red and green lines are hard to see and could pose an issue for color-blind readers.

---

## Referee Comment (RC3) · Anonymous Referee #3 · 30 Oct 2019

General Comments: This paper presents a good overview of the importance of ice charts for not only navigation guidance, but as a relevant archive for climate monitoring and use as a validation variable. This is a very nice paper that provides some good and novel approaches to evaluate how to provide some level of uncertainty in ice charts, particularly with the Kappa statistic and Krippendorff's alpha applications. Additionally, this is hopefully the start of a trend in producing much needed literature that focuses on the role of ice charts in research and development for environmental monitoring. This includes the challenges with how different data sources should be handled and the variation between information input, including the subjective nature of ice analysts. This topic has always been of great interest to the research community that depend

on ice charts as a source for validation and to provide initial conditions in models. With the onset of large volumes of data and predicted increased activity in the Arctic, the ice services are preparing for an integration of automated operational products to assist them in providing accurate information for end-users. For this reason, it is crucial that papers focusing on this topic can clearly communicate 1) why this is important in the operational sense, 2) fundamental challenges in automating sea ice information for operations (which has been going on for the last 30 years) and 3) apply metrics relevant to evaluate automated techniques to the level that can be useful for operations. This paper touches on these three points and opens up more opportunities to explore this topic further in future research.

There are some overall suggestions that would be good to include which would improve the paper. The geophysical limitations with monitoring different sea ice types and concentration relative to the season was mentioned in the conclusion but should be stated earlier in the paper because this is one of the main challenges in sea ice automation. There was no mention of the difference of automation with passive microwave and SAR and seasonal/ regional limitations. Often this has been confused in the research community and should be clear that both sensors will be limited by the same environmental conditions (i.e. wind (noise) and melt). This is will help to clarify the sentence in P3 L29 regarding the melt season caveat.

Specific Comments: Title: Should reflect that this comparison is only a case study using HH polarization in SAR. The current title and the inclusion of passive microwave suggests automation is being applied to various sensors and polarizations. Current classification applications that ice services are evaluating are primarily focused on dual-pol SAR and experimenting with full polarimetry as well.

P2 L3 By stating the analysts visually segment the pixels suggests there is some sort of segmentation application applied here. I am assuming this is referring to how analysts make the decision on where ice and open water is located in the image? If so, state that the ice analyst is able to determine these areas and manually delineates them with

polygons. Then you can omit the statement "(this segmentation is not drawn)".

P2 L6 Here would be a good place to refer to the ice chart manual, MANICE, put out by CIS or the WMO . You can add that reference after this sentence. Please review: https://www.ec.gc.ca/glaces-ice/?lang=En&n=2CE448E2-1&offset=8&toc=show

P2 L28 It would be good to provide a sentence or two on why automating ice information has been a challenge for ice services in the past. We are using the same types of sensors that have been available since passive microwave has been available and with the beginning of the use of SAR in the 1990's. Ice services continue to rely on manually drawn charts because automation for sea ice has significant limitations at the marginal ice zone, coastlines, first year ice types and ice edges for spring and summer sea ice, where we see the greatest amount of traffic in the Arctic.

P3 L20-25: Would be good to include when CIS started to use SAR and the amount of SAR vs PMR that is currently used today. From the understanding in ice services, CIS charts primarily consist of SAR and other high resolution data and only use PMR sparingly because it is unable to detect sea ice features and coastal zones well.

P3 L30-31 Should provide a statement that the current state of automation for ice services will rely on the two channels (dual-pol) until a compact polarimetry is available. Therefore, would be good to provide an explanation as to why was HH only used in this study when the HH/HV has been available with RS2 since 2007.

P4 Figure 1 is misplaced in the manuscript and should be located after its mention in the text.

P4 L2-6 Include reference to MANICE manual and also the Dedrick paper (K.R. Dedrick, K. Partington, M. Van Woert, C.A. Bertoia & D. Benner (2001) U.S. National/Naval Ice Center Digital Sea Ice Data and Climatology, Canadian Journal of Remote Sensing, 27:5, 457-475, DOI: 10.1080/07038992.2001.10854887).

P4 L9 Include reference to WMO manuals 259: JCOMM Expert Team on Sea Ice

(2009) WMO Sea-ice Nomenclature, WMO/OMM/ĐŠĐIJĐđ - No.259 Suppl.No.5. Linguistic equivalents. Geneva, Switzerland, JCOMM Expert Team on Sea Ice, 23pp. (WMO No, 259, Suppl. 5). http://hdl.handle.net/11329/113.

P5 L11 What does "greatest intersectiong overlap" refer to? Please provide a more clear explanation.

P5 L12-16 Where is the comparison done for the Wilcoxon-Mann-Whitney test? Is this a general comparison that had been done before with ice charts or is the image analysis referring to the new polygon generated by an analyst for comparison? Also, and what is "image analysis" referring to? Is this automated image analysis. It appears that you are referring to a previous comparison that had been conducted because you specify the new polygons in this study in the following sentence in P5 L13.

P5 L25-27 Already stated in previous section that only HH is used in this study. Should instead provide reasoning as to why only HH is used when it is first mentioned in P3 L30-31. Again, why is HH only used? You state to "ensure that differences in ice concentration estimates between individuals were restricted to only interpretation of the segmentation, rather than interpretations of the multiple polarizations normally available," however, how does the interpretation of only using one polarization differ from multiple polarzations regarding introducing any bias in the analyst interpretation?

P7 L3-4 We should assume the analyst understands the user interface before doing the assessment. Unless there is something that could be shown with these two polygons that demonstrated the analysts understood the user interface, this disclaimer does not need to be here.

P7 L15-16 One of the major challenges with automating products for operational ice charts are due to the large differences of surface appearances based on regional and seasonal variability. It was noted in the abstract and conclusion but since this is a common problem it would be better to state it in the beginning paragraphs or something that provides information on how monitoring sea ice in these areas vary with respect

to region (fast ice vs. drifting ice) and season, particularly with the melt and Summer season. It will help the reader to have context as to its difficulty and why this data hasn't previously been automated for ice services. Additionally, a table listing the images and dates should be included somewhere so that the reader can get an idea of the types of ice conditions that were being assessed in this study.

P19 L1-6 This paragraph describes the types of images and criteria that were selected for this study. The description of the area selection with regards to the contrast and floe size would be best placed in section 3.1 in order to help set the stage for the study. This section in the conclusion can reiterate and summarize this again and continue to expand on it more detail as you have in the later part of the paragraph.

Technical Comments: P5 L9 "...polygons created by the analyst were compared to the corresponding areas from the published operational charts that used the same Radarsat images."

P5 L11 "The uncertainty of the corresponding coordinate of the polygons show an uneven distribution between one another..."

P13 L4 "The first objective of this study was to compare analyst...."

P13 L6-7 The sentence "Segmentation is not necessarily....." is a very strong statement and could be refuted in some ways without any resources to provide support. You can replace it with something that describes it similar to the following justification: Segmentation papers tend to explore very limited samples of satellite data which they do very well but there are not many papers that apply the same types of techniques across a wider spatial and temporal scales. Whereas ice charts have been produced on a consistent basis for more than 40 years by a wide range of different agencies. Though we know there are differences among ice charts, they overall agree where and what types of ice are present within a given area, with small variations.

P14 L2 Replace "....ranged.." with "...the accepted segmentation results varied between analysts."

P14 L3 Rephrase to state "In 36.8% of total polygons, the analysts were unanimous in agreement with the outcome of the automatic segmentation."

P15 L1-3 "An overall agreement between analyst estimation and segmentation results are shown along the diagonal line, where the proximity of entries outside the line represent the extent to which analysts are over or underestimating ice concentration (Figure 7). There was an overestimation of ice concentration from the analysts with respect to MAGIC.

P15 L4-6 The first sentence is redundant and could flow with the following sentence by combining them and not repeating the same results to state: "Over-estimation of low ice concentrations (i.e. 2/10 to 4/10)resulted in an increase in the number of polygons with high ice concentration (9/10 to 10/10)."

P15 L7-8 Delete "In the cases where some analysts accepted the segmentation results, while others did not, we only considered the responses where it was valid." Already stated in the first sentence in L7.

P15 L9 Delete the parentheses because this is an individual sentence:"Figure 10 shows the combined responses from all participants in this study. (Individual responses are shown in Figure 11)."

P17 L27 Sentence does not need to be put in parentheses.

Figures: Misplaced throughout the manuscript and should be more closely aligned with the text throughout the document. They should be placed after the mention in the text rather than before, or after subsequent figure references.

---

## Author Comment (AC1) · 8 Jan 2020

**The original comment by the reviewer is in black, while our replies are in green. Text directly copied from the original submission is in purple to help facilitate referencing the original submission.**

This paper assesses the accuracy of CIS products and the agreement between analysts in the generation of the CIS ice charts. This type of study is relevant to the field of sea ice mapping since CIS charts can be used as training/validation data for developing automated methods. Thus, it is important to assess the accuracy of these products and understand how they are generated, their strengths and weaknesses.

This paper presents good statistical analyses of the gathered information and is a good start to assess the accuracy of CIS charts. Nonetheless, I would recommend more caution in the discussions section since with more analysts, there could be more of them that disagree with the general consensus than just one out of eight.
We agree. In designing this study, we structured the test such that more analysts can participate and/or more images and polygons could be added to the test dataset, which would help with narrowing the confidence intervals of our estimates.

Does the methodology implemented here reflect how the CIS products are generated? Since CIS offers daily products, they surely refer to previous day analysis to determine ice concentration. This would greatly impact the general agreement between analysts. This should be discussed in section 8.
The methodology used here differs from the method that is used to generate CIS charts. For example, analysts normally have the previous day's chart as a reference. In this study, analysts were asked to estimate ice concentration without access to any other analyst's estimation.

Furthermore, we have added the following text to section 8 of the paper.

"Analysts typically start with the most recent chart when producing a new ice chart. This is done to ensure consistency and continuity between ice charts, and prevent fluctuations and variability in how polygons are drawn, or the information that they contain due to variability in analyst interpretation of SAR. In the past, analysts have carried forward the previous day's ice concentration for a given polygon, unless the analyst estimated what they felt was a significant difference in ice concentration compared to the previous chart. Nowadays, analysts are assigned specific areas to produce charts for over time to ensure better consistency. Therefore, ice concentrations in charts may exhibit higher agreement between analysts' estimates than they do without a reference chart. In a future study, analysts could be given an ice concentration estimate and then asked if they agreed or disagreed with the estimate to see if the result changes based on what the estimate given was."

Another analysis that I should be added to this study, and the authors partially addressed it in section 8, is how the agreements vary with ice types and not only ice concentrations. Ice type is also a significant variable for shipping and weather/climatology modeling. I suggest adding details

in section 3 about the distribution of ice concentrations and ice types for all the different polygons to show how this study reflects the ice conditions seen in Canadian waters.
We propose adding this text to section 3.

"The images used for this study were selectively picked to be areas with well-defined floes with high contrast against the black (water) background. SAR image quality varies from image to image, and even within image. Likewise, the structure of sea ice in Canadian waters can vary greatly, with brash and rubble ice along the East Coast and well-defined floes in the Beaufort Sea. Ice without well-defined spatial structure may not be captured due to the resolution of the sensor. For example, first year ice can appear similar to open water, making it difficult to determine its edges. Brash ice is composed of small pieces of ice (less than 2 meters in diameter) that cannot be resolved at the resolution of the (SAR) sensor. Furthermore, segmentation of sea ice in visually ambiguous conditions (i.e. first year ice during the melt season; brash ice; etc.) by automated algorithms is still sub-optimal. As a result, we did not present analysts with ice conditions that would have been difficult to automatically segment. The sea ice types used in the samples of this study are not representative of all sea ice conditions typically found in Canadian Service Ice Charts. This study quantifies the accuracy of sea ice concentration estimates under the best case scenario of well-defined floes in very clear SAR images. It is expected that accuracy would decrease under brash ice conditions and/or poor image quality."

I think in a future study, the authors should look at how MAGIC compared to CIS charts generated operationally to assess the agreement between both products with a higher statistical significance than just imagery with low contrast. The authors should elaborate a bit more in the discussion section how MAGIC performs in low contrast or different ice conditions and how this could impact results of this study.
We restricted our analysis to only instances with high contrast to measure analyst ability to estimate sea ice concentration as our primary focus, rather than assessing the performance of MAGIC. However, we used the results of the analysis to make some general comments about MAGIC. We agree that comparison against MAGIC for a future study would be beneficial to compare how MAGIC compared to sea ice charts in general, without being validated by any analysts.

P.4. , L10. Add "(floe size)" after "predominant form (F) of ice"
We have changed the document as suggested.

P.16, L33. I imagine that 7/10 is considered 1/10 overestimation in this case? Please clarify.
That's the correct interpretation. The sentence has now been changed to "For example, if a polygon had the modes 5/10 and 6/10, then an ice concentration estimate of 4/10 was considered 1/10 under-estimation away from 5/10 and 7/10 was considered 1/10 over-estimation."

---

## Author Comment (AC2) · 8 Jan 2020

**Reply to Reviewer #2**

**The original comment by the reviewer is in black, while our replies are in green. Text directly copied from the original submission is in purple to help facilitate referencing the original submission.**

The paper presents a comparison of ice analyst assessments of sea ice concentration values at the Canadian Ice Service, both between analysts and against an automated retrieval method. It is well written and I see only some minor issues that can be fixed or explained better. The study is limited to Canadian Ice Service analysts, and finds very good agreement between them. With more analysts, including from other ice services, it is likely the range of opinions will increase. However running this type of exercise is time intensive, so only a limited number of such studies have been per-formed (i.e. Karvonen et al, 2015). Other studies have tried examining the areas where ice charts overlap, for example a comparison between Norwegian and Danish ice charts at https://www.ecmwf.int/sites/default/files/elibrary/2018/17968-satellite-observations-sea-ice-concentration-and-drift.pdf(p.14-15). More of this type of paper evaluating ice chart data are needed to provide quantitative measures of uncertainty that can be used when ice charts are used as training data for automated classification algorithm developments and as climate records.

We were quite fortunate to have eight analysts participate in this study and agree that there is difficulty in finding operational analysts available to participate in research studies, which is likely why there are few studies in this area. We also agree that more papers quantifying uncertainty in ice charts would be useful and are hopeful to continue to contribute to this area of research.

P2, L2. Analysts at the CIS delineate boundaries around regions with similar ice conditions. This raises the question, what was the rationale for the example polygon areas in Figures 3 and 4 as these contain dissimilar ice conditions?

We used different criteria to delineate sample polygons than the criteria normally used to delineate polygons in regular ice charts. Emphasis was placed on identifying areas with fractional ice cover, since there is little value in evaluating analysts' ability to estimate 0/10 or 10/10 ice concentration. Secondly, polygons were not drawn with the purpose of classifying the entire image; instead, polygons were only drawn in areas with fractional ice cover and high contrast between ice and open water to maximize the automated algorithm's ability to clearly demarcate ice from open water.

P2 L28-34. This uncertainty is not limited to ice charts, automatic satellite derived sea ice concentration products also have varying levels of uncertainty.

The specific lines referred to are:

"The uncertainty of sea ice concentration estimates can result in downstream uncertainties for applications that rely on sea ice charts. For example, sea ice concentration estimates from Canadian Ice Service charts are used as a data source for input to initialize sea ice models (Smith et al., 2015; Lemieux et al, 2015). The error in the initial condition of sea ice concentration estimates can propagate and grow with time, and impact the accuracy of prediction from numerical models (Parkinson et al., 2001). Uncertainty of ice concentration estimates could also impact the accuracy of climatology studies of ice concentration derived from operational ice charts, although that has not been investigated."

Agreed; satellite derived products also have uncertainty. We tried to minimize the uncertainty from automated satellite derived products in this study by having analysts validate the output from the segmentation.

P3 L31 and P5 L25
How does limiting the analysts to only HH affect their choice of concentration? As noted at L29, the addition of HV has improved distinguishing between ice and water, and Canadian analysts have had access to this form of RADARSAT-2 for over a decade. One would imagine limiting to HH-only would be generating additional analyst uncertainty. It would be interesting in a further study to provide one set of images as HH, and another as HV.
Typically, ice charting is done with HH as a primary polarization, and HV is only used to distinguish ambiguous ice types. However, the sample polygons used in this study focused on examples with minimal ambiguity.

We have added the following text to the paper to clarify why only HH was used: "Only the HH band was used for both segmentation and visual interpretation in this study. HH was the only polarization available in samples drawn in RADARSAT-1 imagery, and therefore, was the only band used for both segmentation and visual interpretation. HH was the only polarization used for samples drawn in RADARSAT-2 imagery as well, to be consistent with the polarizations available for all images. Typically, ice charting is done with HH as a primary polarization, and HV is only used to distinguish ambiguous ice types. However, the sample polygons used in this study focused on examples with minimal ambiguity."

P4, L8. The egg code is a World Meteorological Organization international standard. Reference should be made to WMO publication 259, Sea Ice Nomenclature.
Agreed. The WMO publication has been referenced in the paper.

P6 Figure 3
The example provided shows an area containing both dark areas of open water and lighter areas of concentrated sea ice. The open water areas are larger and contiguous enough that most ice analysts would delineate them as separate areas, and some explanation is needed as to why these should be grouped with large areas of concentrated ice. MAGIC looks to have done a good job of marking these open water areas.
The purpose of the study was to assess the agreement in ice concentration estimates of analysts with a polygon. The purpose of the study was not to assess the way that polygons were drawn (although we would like to pursue this in a future study!)

P8 Figure 4. Similar question, why does the polygon contain areas of areas of very open drift ice that are continuous with adjacent areas outside the polygon?
Homogeneity of ice was not a requirement in our study. Our only requirement was defining areas of high contrast between ice and open water to ensure there was little ambiguity in differentiating between them.

P13 L11-P14 L3.   181 disagreements, or almost a third, is a large proportion.  MAGIC provided lower estimates in 175 of cases, but analysts tend to err on the conservative side, and if the mix of ice concentrations shown in the example image polygons is representative of the dataset, it is expected that there would be more disagreement, as shown by only 36.8% of examples having unanimous agreement between analysts.

It was stated in the abstract that "true accuracy is expected to be lower than what is found in this study," because we restricted our samples to only those cases where ice and water had high separability.  Satellite images often have areas of ambiguity in separating ice from open water, where we expect MAGIC to have greater difficulty.  Therefore, we expect true accuracy to be lower than what we found in this study.

P18 L31-32.  A 84% at plus/minus one-tenth accuracy shows very good analyst skill, most ice services only work in concentration ranges.

Reply to reviewer:  I have now added this information to the paper.  "Many chart products by Ice Services report ice concentration in ranges, rather than specific tenths; therefore, 84% shows good analyst skill."

P19 L22-23.  Agree, the definition of the polygon is critical.  The examples provided appear somewhat ambiguous so some explanation as to why the areas shown in the examples are as they are would be useful.

We used different criteria to delineate sample polygons than the criteria normally used to delineate polygons in regular ice charts.  Emphasis was placed on identifying areas with fractional ice cover, since there is little value in evaluating analysts' ability to estimate 0/10 or 10/10 ice concentration.  Secondly, polygons were not drawn with the purpose of classifying the entire image; instead, polygons were only drawn in areas with fractional ice cover and high contrast between ice and open water to maximize the automated algorithm's ability to clearly demarcate ice from open water.

P19, L35 – P20, L1.  The ice charts are for maritime safety so will err on the cautious side.

Reply to reviewer:  Agreed, and the sentence is modified as follows.  "This could result in ships being prevented from going into areas that they normally would be able to enter; however, Ice Service charts are produced for maritime safety, and therefore, err on the cautious side."

Technical Corrections

Figure 2.  Difficult to view at this size and could be larger.
We have resized the figure to be larger.

Figures 2 and 4. The red and green lines are hard to see and could pose an issue for color-blind readers.
We have changed the colour to purple and yellow.

---

## Author Comment (AC3) · 8 Jan 2020

**Reply to Reviewer #3**

**The original comment by the reviewer is in black, while our replies are in green. Text directly copied from the original submission is in purple to help facilitate referencing the original submission.**

General Comments: This paper presents a good overview of the importance of ice charts for not only navigation guidance, but as a relevant archive for climate monitoring and use as a validation variable. This is a very nice paper that provides some good and novel approaches to evaluate how to provide some level of uncertainty in ice charts, particularly with the Kappa statistic and Krippendorff's alpha applications. Additionally, this is hopefully the start of a trend in producing much needed literature that focuses on the role of ice charts in research and development for environmental monitoring. This includes the challenges with how different data sources should be handled and the variation between information input, including the subjective nature of ice analysts. This topic has always been of great interest to the research community that depend on ice charts as a source for validation and to provide initial conditions in models. With the onset of large volumes of data and predicted increased activity in the Arctic ,the ice services are preparing for an integration of automated operational products to assist them in providing accurate information for end-users. For this reason, it is crucial that papers focusing on this topic can clearly communicate 1) why this is important in the operational sense, 2) fundamental challenges in automating sea ice information for operations (which has been going on for the last 30 years) and 3) apply metrics relevant to evaluate automated techniques to the level that can be useful for operations. This paper touches on these three points and opens up more opportunities to explore this topic further in future research.

We agree with many of the points the reviewer has made above regarding the relevancy of this area of research and current challenges that we face.

There are some overall suggestions that would be good to include which would improve the paper. The geophysical limitations with monitoring different sea ice types and concentration relative to the season was mentioned in the conclusion but should be stated earlier in the paper because this is one of the main challenges in sea ice automation. There was no mention of the difference of automation with passive microwave and SAR and seasonal/ regional limitations. Often this has been confused in the research community and should be clear that both sensors will be limited by the same environmental conditions (i.e. wind (noise) and melt). This is will help to clarify the sentence in P3 L29 regarding the melt season caveat.

We would like to thank the reviewer for making overarching suggestions for placing this paper within the context of automated detection algorithms (including current difficulties and why this has not already been implemented across Ice Services). Our original submission discussed this much more briefly.

We have modified the paper to address the difficulty of monitoring different types of sea ice earlier, as suggested. This helped to explain how the polygons were delineated for this study as well. We address this point further below when addressing the specific comments by the reviewer.

Title should reflect that this comparison is only a case study using HH polarization in SAR. The current title and the inclusion of passive microwave suggests automation is being applied to various sensors and polarizations. Current classification applications that ice services are evaluating are primarily focused on dual-pol SAR and experimenting with full polarimetry as well. The title of the paper has been modified to show that only HH was used in this study.

P2, L3. By stating the analysts visually segment the pixels suggests there is some sort of segmentation application applied here. I am assuming this is referring to how analysts make the decision on where ice and open water is located in the image? If so, state that the ice analyst is able to determine these areas and manually delineates them with polygons. Then you can omit the statement ("this segmentation is not drawn").

The sentences in question: "Analysts at the CIS delineate boundaries around regions with similar ice conditions, for navigational purposes (we refer to these as polygons in the following). Next, they visually segment the pixels inside the polygon into ice or water pixels (this segmentation is not drawn). The analyst then assigns an estimated concentration value for the polygon using the visual segmentation".

Correct… We have changed the sentence to "Analysts at the CIS identify areas with similar ice conditions and open water for navigational purposes, then manually delineate them with polygons. The analyst then assigns an estimated concentration value for the polygon."

P2, L6. Here would be a good place to refer to the ice chart manual, MANICE, put out by CIS or the WMO. You can add that reference after this sentence. Please review: https://www.ec.gc.ca/glaces-ice/?lang=En&n=2CE448E2-1&offset=8&toc=show A citation has been added for MANICE when describing types of ice charts produced at the Canadian Ice Service.

P2, L28. It would be good to provide a sentence or two on why automating ice information has been a challenge for ice services in the past. We are using the same types of sensors that have been available since passive microwave has been available and with the beginning of the use of SAR in the 1990's. Ice services continue to rely on manually drawn charts because automation for sea ice has significant limitations at the marginal ice zone, coastlines, first year ice types and ice edges for spring and summer sea ice, where we see the greatest amount of traffic in the Arctic.

Section 2.1 was greatly revised to address and expand on these issues. Please refer to the revised document for this section.

P3, L20-25. Would be good to include when CIS started to use SAR and the amount of SAR vs. PMR that is currently used today. From the understanding in ice services, CIS charts primarily consist of SAR and other high resolution data and only use PMR sparingly because it is unable to detect sea ice features and coastal zones well.

We have added this paragraph to section 2.1. Specifically, we added the following paragraph:

"The CIS relied on RADARSAT-1, a SAR sensor, for ice charting beginning in 1996 until its decommissioning in 2013. The Canadian Ice Service currently relies predominantly on

RADARSAT-2, but will start to use the RADARSAT Constellation Mission (RCM) operationally, following its recent launch in 2019. In the 2017 calendar year, the Canadian Ice Service received approximately 45 000 SAR scenes between Sentinel-1 and RADARSAT-2, and another 85 000 scenes from various satellites including GOES, MODIS, AMSR, and VIIRS. The lower number of SAR scenes reflects the fact that RADARSAT scenes are geographically targeted acquisitions ordered by the CIS, while GOES, MODIS, AMSR and VIIRS are publicly available swaths acquired for general use. The latter are less targeted for CIS Operations, but useful as a secondary, supplemental data source."

P3, L30-31. Should provide a statement that the current state of automation for ice services will rely on the two channels (dual-pol) until a compact polarimetry is available. Therefore, would be good to provide an explanation as to why HH only used in this study when the HH/HV has been available with RS2 since 2007.
Regarding dual-polarization, we added the following in section 2.1

"Automation of sea ice classification algorithms currently use dual-polarization imagery, but will use compact polarimetry as it becomes available."

With respect to why only HH was used, we added the following text in section 3.2.

"Only the HH band was used for both segmentation and visual interpretation in this study. Typically, ice charting is done with HH as a primary polarization, and HV is only used to distinguish ambiguous ice types. However, the sample polygons used in this study focused on examples with minimal ambiguity."

P4. Figure 1 is misplaced in the manuscript and should be located after its mention in the text.
All figures have been moved to the end of the document (as it was supposed to be according to the template).

P4 L2-6 . Include reference to MANICE manual and also the Dedrick paper (K.R. Dedrick, K. Partington, M. Van Woert, C.A. Bertoia & D. Benner (2001) U.S. National/Naval Ice Center Digital Sea ice Data and Climatology, Canadian Journal of Remote Sensing, 27:5, 457-475.
Reference has been added to the paper.

P4 L9 Include reference to WMO manuals 259: JCOMM Expert Team on Sea Ice (2009) WMO Sea-ice Nomenclature, WMO/OMM/ĐŠĐIJĐ ̄đ - No.259 Suppl.No.5. Lin-guistic equivalents. Geneva, Switzerland, JCOMM Expert Team on Sea Ice, 23pp.(WMO No, 259, Suppl. 5). http://hdl.handle.net/11329/113.
We have added this reference to the paper.

P5, L11. What does "greatest intersecting overlap" refer to? Please provide a more clear explanation.

The sentence in question. "The polygon sizes were compared to polygon sizes from the published operational daily charts and image analyses that used the same RADARSAT images. The sample polygon sizes were compared to the sizes of polygons from published charts with the greatest intersecting overlap."

We tried to find a polygon from the published ice charts that corresponded to our samples. Since our sample polygons sometimes spatially overlapped more than one polygon, we took the one with the greatest intersecting overlap.

We have changed the phrasing in the paper to be:
"Since the polygons were delineated differently, sometimes the sample polygon would spatially intersect with two or more polygons; making it difficult to directly compare the sizes of polygons. We addressed this by identifying the polygon with the greatest spatial intersection with the sample polygon, and comparing the two areas."

P5, L12-16. Where is the comparison done for the Wilcoxon-Mann-Whitney test? Is this a general comparison that had been done before with ice charts or is the image analysis referring to the new polygon generated by an analyst for comparison? Also, and what is "image analysis" referring to? Is this automated image analysis. It appears that you are referring to a previous comparison that had been conducted because you specify the new polygons in this study in the following sentence in P5 L13.

The sentences in question: "The polygon sizes were compared to polygon sizes from the published operational daily charts and image analyses that used the same RADARSAT images."

We believe there is some confusion as to the difference between image analysis charts, daily charts, and the samples generated for our study. We have since added a new citation for MANICE, which explains what an image analysis is.

We found polygons from published image analysis charts and daily charts, then compared the sizes of those polygons to the polygon samples in our study to assess if the sample polygons' sizes greatly different from the sizes of polygons from image analyses and dailies.

We have changed the text as follows. "The polygon sizes were compared to polygon sizes from two types of published operational charts: daily charts and image analyses. The image analyses and daily charts used the same RADARSAT images that were used to delineate polygons used in this study. Of interest was determining if there were differences in the size of polygons drawn for this study and the sizes of polygons in published charts, since polygon sizes could impact analyst ability to estimate ice concentration."

P5, L25-27. Already stated in previous section that only HH is used in this study. Should instead provide reasoning as to why only HH is used when it is first mentioned in P3 L30-31. Again, why is HH only used? You state to "ensure only difference in ice concentration estimates between individuals were restricted to only interpretation of the segmentation, rather than interpretations of

the multiple polarizations normally available," however, how does the interpretation of only using one polarization differ from multiple polarizations regarding introducing any bias in the analyst interpretation?

We have revised section 3.2 to explain why only HH was used.

"Only the HH band was used for both segmentation and visual interpretation in this study. Typically, ice charting is done with HH as a primary polarization, and HV is only used to distinguish ambiguous ice types. However, the sample polygons used in this study focused on examples with minimal ambiguity."

Since we selected only samples with high separability between ice and open water, there was very little ambiguity. HV is typically used for differentiating sea ice in difficult conditions, which was not the case in our study. Therefore, only HH was used.

P7, L3-4. We should assume the analyst understands the user interface before doing the assessment. Unless there is something that could be shown with these two polygons that demonstrated the analysts understood the user interface, this disclaimer does not need to be here.

Although we agree, we felt this was worth addressing for the Operational folks who would question whether some of the results were due to lack of understanding. Also, we felt it was worth mentioning because the analysts were initially confused by the way the data was being presented to them. This may be useful information for others trying to replicate a study of this type.

P7, L15-16.

One of the major challenges with automating products for operational ice charts are due to the large differences of surface appearances based on regional and seasonal variability. It was noted in the abstract and conclusion but since this is a common problem it would be better to state it in the beginning paragraphs or something that provides information on how monitoring sea ice in these areas vary with respect to region (fast ice vs. drifnt ice) and season, particularly with the melt and Summer season. It will help the reader to have context as to its difficulty and why this data hasn't previously been automated for ice services. Additionally, a table listing the images and dates should be included somewhere so that the reader can get an idea of the types of ice conditions that were being assessed in this study.

With respect to the first part of this comment, 2.1 has been revised to contain information about automated classification of sea ice in satellite imagery, with some examples of difficult conditions for automatic classification.

A table has been added to the appendix listing the image acquisition file names and date of acquisition.

P19 L1-6. This paragraph describes the types of images and criteria that were selected for this study. The description of the area selection with regards to the contrast and floe size would be best placed in section 3.1 in order to help set the stage for the study. This section in the conclusion can reiterate and summarize this again and continue to expand on it more detail as you have in the later part of the paragraph.

Agreed; this paragraph was moved to 3.1. Judging by the reviewers' comments, there was definitely a lack of clarity in how the polygons were drawn so it makes little sense that this explanation was given at the end, in the conclusions.

Technical Comments

P5 L9. "…polygons created by the analyst were compared to the corresponding areas from the published operational charts that used the same Radarsat images."
This is better phrasing but the wording was already addressed/changed with a previous revision: "Since the polygons were delineated differently, sometimes the sample polygon would spatially intersect with two or more polygons; making it difficult to directly compare the sizes of polygons. We addressed this by identifying the polygon with the greatest spatial intersection with the sample polygon, and comparing the two areas."

P13, L4. "The first objective of this study was to compare analyst…"
Original sentence: "This part of the study focuses on the first objective of this study, which was to compare analyst…"
We agree with the correction.

P13, L6-7. The sentence "Segmentation is not necessarily…" is a very strong statement and could be refuted in some ways without any resources to provide support. You can replace it with something that describes it similar to the following justification: Segmentation papers tend to explore very limited samples of satellite data which they do very well but there are not many papers that apply the same types of techniques across a wider spatial and temporal scales. Whereas ice charts have been produced on a consistent basis for more than 40 years by a wide range of different agencies. Though we know there are differences among ice charts, they overall agree where and what types of ice are present within a given area, with small variations."
We replaced the text with your suggestion accordingly. "Automated sea ice classification algorithms often use sea ice charts as a truth dataset for verification since they cover large geographic areas and have been produced for many years by many Ice Services. Furthermore, while there are differences among ice charts, they generally agree with respect to types of ice present and where they occur."

P14, L2. Replace "…ranged…" with "…the accepted segmentation results varied between analysts."
The original sentence in question: "Furthermore, analysts ranged in their level of acceptance of the segmentation results."
We agree with the correction.

P14, L3. Rephrase to state "In 36.8% of total polygons, the analysts were unanimous in agreement with the outcome of the automatic segmentation."
The original sentence in question: "In 36.8% of the total polygons, all analysts unanimously stated they accepted the segmentation results."
Changed; we agree that the reviewer #3's phrasing was superior.

P15 L1-3.  "An overall agreement between analyst estimation and segmentation results are shown along the diagonal line, where the proximity of entries outside the line represent the extent to which analysts are over or underestimating ice concentration (Figure 7).  There was an overestimation of ice concentration from the analysts with respect to MAGIC."

The original sentence in question.  "Perfect agreement between analyst estimation and the segmentation results lie along the diagonal; entries below (above) the diagonal show over (under) estimation by analysts: the analysts tend to over-estimate the ice category with respect to MAGIC."

Changed; we agree that the reviewer #3's phrasing was superior.

P15, L4-6.  The first sentence is redundant and could flow with the following sentence by combining them and not repeating the same results to state:  "Over-estimation of low ice concentration (i.e. 2/10 to 4/10) resulted in an increase in the number of polygons with high ice concentration (9/10 to 10/10)."

Changed to the reviewer's suggestion.

P15, L7-8.  Delete "In the cases where some analysts accepted the segmentation results, while others did not, we only considered the responses where it was valid."  Already stated in the first sentence in L7.

Changed to the reviewer's suggestion.

P15, L9.  Delete the parentheses because this is an individual sentence:  "Figure 10 shows the combined responses from all participants in this study.  (Individual responses are shown in Figure 11)."

Changed; we agree.

P17, L27.  Sentence does not need to be put in parentheses.

Changed, agreed.

Misplaced throughout the manuscript and should be more closely aligned with the text throughout the document.  They should be placed after the mention in the text rather than before, or after subsequent figure references.

All figures were moved to the end of the document (as it was supposed to be according to the template).